# Mutant CEBPA promotes tolerance to inflammatory stress through deficient AP-1 activation

Maria Cadefau-Fabregat [1,2,3], Gerard Martínez-Cebrián[1,11], Lucía Lorenzi [1,11], Felix D. Weiss [4], Anne-Katrine Frank[5,6], José Manuel Castelló-García[7], Eric Julià-Vilella[1,3], Andrés Gámez-García [7], Laura Yera[1], Carini Picardi Morais de Castro[1,3], Yi-Fang Wang[8,9], Felix Meissner [4], Alejandro Vaquero [7], Matthias Merkenschlager [8,9], Bo T. Porse [5,6,10] & Sergi Cuartero [1,2] ✉

The CEBPA transcription factor is frequently mutated in acute myeloid leukemia (AML). Mutations in the *CEBPA* gene, which are typically biallelic, result in the production of a shorter isoform known as p30. Both the canonical 42-kDa isoform (p42) and the AML-associated p30 isoform bind chromatin and activate transcription, but the specific transcriptional programs controlled by each protein and how they are linked to a selective advantage in AML is not well understood. Here, we show that cells expressing the AML-associated p30 have reduced baseline inflammatory gene expression and display altered dynamics of transcriptional induction in response to LPS, consequently impacting cytokine secretion. This confers p30-expressing cells an increased resistance to the adverse effects of prolonged exposure to inflammatory signals. Mechanistically, we show that these differences primarily arise from the differential regulation of AP-1 family proteins. In addition, we find that the impaired function of the AP-1 member ATF4 in p30-expressing cells alters their response to ER stress. Collectively, these findings uncover a link between mutant CEBPA, inflammation and the stress response, potentially revealing a vulnerability in AML.

Lineage-specific transcription factors (TF) play a pivotal role in orchestrating the molecular control of cell identity and multicellularity[1–4]. TFs recognize specific DNA motifs, which cluster into small regulatory regions such as enhancers and promoters to modulate gene transcription[1,5,6]. The aggregation of multiple TF binding sites within regulatory elements enables complex combinatorial dynamics—including cooperation[7–9], antagonism[10–12], or redistribution[13–15]—further increasing the specificity and precision of transcriptional control. TF network dynamics rely on the fine-

[1]Josep Carreras Leukaemia Research Institute (IJC), Badalona, Spain. [2]Germans Trias i Pujol Research Institute (IGTP), Badalona, Spain. [3]Doctoral Program in Biomedicine, Universitat de Barcelona (UB), Barcelona, Spain. [4]Institute of Innate Immunity, Department for Systems Immunology and Proteomics, Medical Faculty, University Hospital Bonn, University of Bonn, 53127 Bonn, Germany. [5]The Finsen Laboratory, Copenhagen University Hospital—Rigshospitalet, Copenhagen, Denmark. [6]Biotech Research and Innovation Centre (BRIC), Faculty of Health Sciences, University of Copenhagen, Copenhagen, Denmark. [7]Chromatin Biology Laboratory, Josep Carreras Leukaemia Research Institute (IJC), Badalona, Spain. [8]MRC London Institute of Medical Sciences, Institute of Clinical Sciences, Faculty of Medicine, Imperial College London, Du Cane Road, London W12 0NN, UK. [9]Institute of Clinical Sciences, Faculty of Medicine, Imperial College London, Du Cane Road, London W12 0NN, UK. [10]Department of Clinical Medicine, University of Copenhagen, Copenhagen, Denmark. [11]These authors contributed equally: Gerard Martínez-Cebrián, Lucía Lorenzi. ✉e-mail: scuartero@carrerasresearch.org

tuned quantitative balance of expressed TFs, which is intricately linked to the translational control of isoform expression[16,17]. Despite the robustness conferred by TF regulatory networks[18,19], perturbations in specific TFs can alter gene expression programs and lead to developmental defects[19], autoimmune disorders[20], and cancer[21].

The bZIP transcription factor CEBPA is frequently mutated (4–12%) in acute myeloid leukemia (AML)[22–24]. It is increasingly expressed from hematopoietic stem and progenitor cells (HSPCs) to terminally differentiated myeloid cells such as granulocytes and macrophages, and its absence results in arrested myeloid differentiation at the common myeloid progenitor (CMP) stage[25–27]. Two CEBPA isoforms can be translated from the same transcript using different translation start sites, a canonical 42-kDa protein (p42) and a shorter 30-kDa isoform (p30), that lacks two trans-activation domains at the N-terminal region[28,29]. In AML, *CEBPA* mutations can be grouped into two main classes: frameshift mutations at the N-terminal region of the gene result in the exclusive translation of the short isoform p30, while in-frame mutations within the C-terminal bZIP domain compromise DNA binding and dimerization[30,31]. Typically, patients accumulate both classes of mutations in a biallelic manner[24,32,33], and therefore mutant cells only express one CEBPA isoform (p30) with an intact bZIP domain. However, the regulatory properties of the p30 isoform compared to the canonical long isoform (p42) remain incompletely understood[27,34–37], precluding new insights into the selective advantage of *CEBPA*-mutant clones and the identification of therapeutic strategies for precision medicine.

While both isoforms retain the capacity to bind the same DNA motif and their genomic distribution is highly similar[35,37], hematopoietic progenitors exclusively expressing p30 display increased self-renewal and form leukemia-initiating myeloid blasts in vivo[38,39]. Moreover, p30 can selectively activate transcription of specific genes such as *Msi2*[37], *Nt5e*[35] and *Gata2*[40] in an isoform-specific manner, and has isoform-specific chromatin interactors[41,42]. However, it remains unclear how the differential regulation of gene expression by p30 confers a selective advantage to *CEBPA*-mutant AML. Here, we set out to identify new transcriptional programs differentially controlled by CEBPA isoforms and to delineate the molecular mechanisms underlying the different transcriptional output. We show that p30 is unable to promote normal expression of inflammatory pathways, which leads to a reduced capacity to mount a proper inflammatory response to LPS. As persistent, chronic inflammatory stimuli eventually result in reduced fitness and apoptosis[43,44], p30-expressing cells become partially protected and gain fitness advantage. Mechanistically, we show that this is partially due to a reduced capacity by the p30 isoform to transcriptionally activate AP-1 members such as *Fos*, as well as its incapacity to directly interact with another stress responsive AP-1 member, ATF4. Finally, we show that the altered function of ATF4 may represent a specific vulnerability of *CEBPA*-mutant cells.

## Results

### The p30 isoform is associated with decreased inflammatory gene expression

To define the transcriptional impact of p30 and p42 isoforms on HSPCs, we first used the mouse hematopoietic progenitor cell line HPC-7[45]. HPC-7 cells recapitulate the transcriptomic and epigenetic landscape of mouse HSCs[46,47]; do not harbor typical AML mutations; and exhibit very low levels of endogenous CEBPA (Supplementary Fig. 1a), making them an ideal model to delineate the transcriptomic effects of p30 and p42 in HSCs. We stably transduced HPC-7 cells with plasmids constitutively expressing either p30 or p42. To prevent premature differentiation driven by CEBPA expression, the transgenes were fused to an ERT2 fragment, allowing for the control of nuclear entry with 4-hydroxytamoxifen (4-OHT) (Supplementary Fig. 1b). As a result, each cell line only expressed the ectopic isoform without significant upregulation of endogenous CEBPA (Supplementary Fig. 1a)

and cells did not show signs of premature myeloid differentiation (Supplementary Fig. 1c, d), despite minor leakiness prior to 4-OHT addition (Supplementary Fig. 1b).

We conducted RNA sequencing (RNA-seq) analysis after 24 h of 4-OHT treatment. Compared to control cells transduced with an empty vector, expression of p30 and p42 resulted in the up- and down-regulation of hundreds of genes, and the majority of changes were isoform-specific (Supplementary Fig. 1e), which is consistent with previous reports[35,37,40] and underscores the capacity of CEBPA to regulate transcription in an isoform-specific manner. To directly identify which genes are differentially controlled by the two isoforms, we compared p30 and p42-expressing cells directly and identified 709 and 822 upregulated and downregulated genes, respectively (Fig. 1a) and performed gene ontology (GO) analysis. Surprisingly, the top ten gene ontology terms enriched among downregulated genes in p30 cells were all related to inflammatory gene expression, including 'regulation of inflammatory response', 'regulation of cytokine production' and 'defense response to bacterium' (Fig. 1b). In line with this, gene set enrichment analysis (GSEA) showed that inflammatory genes were downregulated in p30 cells (NES = –1.82, FDR = 0.001, Supplementary Fig. 1f). This finding was unexpected as there has been no prior indication of different contributions of these two isoforms to the inflammatory response. To rule out the possibility that these differences were caused by tamoxifen-induced effects, we repeated the experiment in HPC-7 cells expressing non-fused, constitutive forms of p30 and p42. RNA-seq analysis of these cells confirmed a similar downregulation of inflammatory genes (Supplementary Fig. 1g-j), indicating that the effect is driven by the isoforms themselves rather than by any artifact of the fusion proteins.

To confirm these effects in primary cells, we next analyzed mouse primary hematopoietic progenitor cells. Since most p30 homozygous mice die before birth[38], we isolated hematopoietic progenitors from the bone marrow of *Cebpa*[Fl/p30]; *R26-Cre-ER* mice. To promote the exclusive expression of p30 in these progenitor cells, the floxed wild-type (WT) allele was deleted within 72 h of inducible Cre nuclear translocation by 4-OHT (Supplementary Fig. 1k, l). We then used RNA-seq to profile gene expression of floxed p30/- cells compared to 4-OHT-treated WT controls, which predominantly express the p42 allele[35,38]. We performed gene ontology analysis of the 648 downregulated genes, once again showing a majority of inflammatory gene sets among the most significantly downregulated terms (Fig. 1c, d). Most of the deregulated genes in progenitor cells that were also affected in HPC-7 showed changes in the same direction, indicating commonalities between the two systems (Supplementary Table 1).

Since expression of p30 and p42 in primary progenitors could potentially be skewing myeloid differentiation trajectories, differences in progenitor population composition or differentiation pace may explain the observed variations in inflammation-related gene expression in bulk RNA-seq. To address this possibility, we examined if the reduced inflammatory gene expression was also evident when comparing populations at the same differentiation stage. For this, bone marrow cells from *Cebpa*[Fl/p30]; *R26-Cre-ER* and control mice were cultured in macrophage differentiation media without immediately deleting the p42 allele. Instead, 4-OHT was added at 96 hours, ensuring a normal differentiation history before p42 deletion. This strategy enabled the comparison of equivalent populations of fully differentiated p30 and WT macrophages, without differences in cell maturity (Supplementary Fig. 1m-p). Gene expression profiling by RNA-seq confirmed the reduction of baseline inflammatory gene expression in p30 macrophages (Fig. 1e, f). Thus, the like-for-like comparison between control and p30-expressing macrophages ruled out the possibility that the reduced inflammatory transcription was related to impaired differentiation of p30 cells.

In summary, the three experimental systems (HPC-7 cells, primary hematopoietic progenitors, and primary macrophages) showed highly

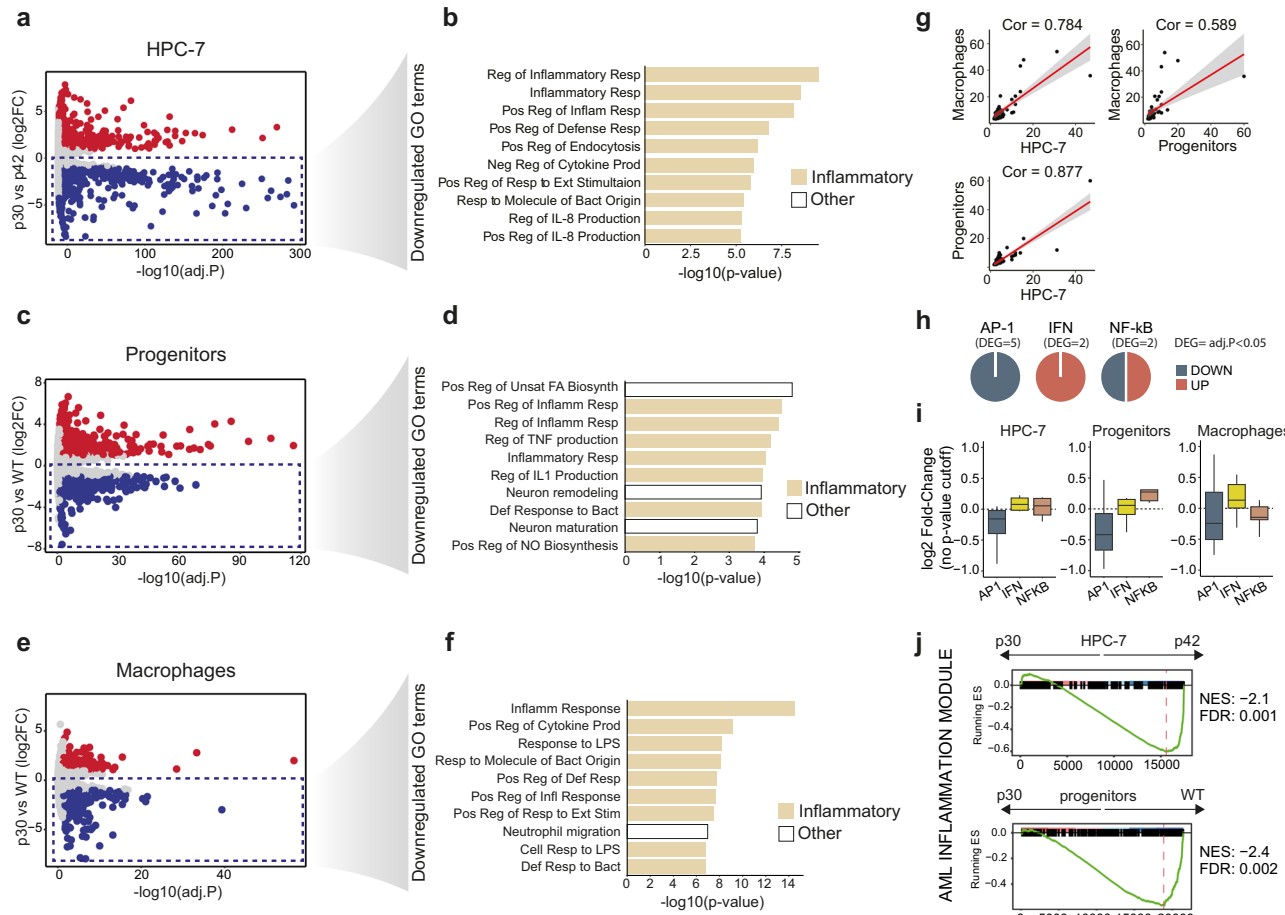

**Fig. 1 | Downregulation of inflammatory genes in p30-expressing mouse HSCs, progenitors and macrophages.** Volcano plot of differentially expressed genes comparing p30 vs p42 in (**a**) HPC-7 cells, (**c**) primary hematopoietic progenitors and (**e**) macrophages. Red: significantly upregulated genes (adj.P < 0.05, log2FC > 1), blue: significantly downregulated ones (adj.P < 0.05, log2FC < -1). Top 10 gene ontology (GO) terms enriched in downregulated genes in (**b**) HPC-7 cells, (**d**) hematopoietic progenitors and (**f**) macrophages. **g** Correlation of enriched GO terms between macrophages, HPC-7, and bone marrow hematopoietic progenitors. Axes show the odds ratio of enrichment of each term in the indicated cell type. Red line: Linear regression and standard error (gray). **h** Fraction of significantly up- and downregulated AP-1/IFN/NFκB transcription factor family genes in HPC-7 p30 vs p42 (only genes with adj.P < 0.05 are shown). **i** Fold-change (log2) in expression of p30

cells of AP1/IFN/NFκB regulators (no adj.P cutoff), in HPC-7 (left), hematopoietic progenitors (middle) and macrophages (right). Center line represents the median, lower and upper hinges correspond to the first and third quartiles (25th and 75th percentiles). Upper whisker extends from the hinge to the largest value no further than 1.5 * IQR from the hinge (IQR, inter-quartile range). Lower whisker extends from the hinge to the smallest value at most 1.5 * IQR of the hinge. **j** GSEA of HPC-7 and hematopoietic progenitors of an AML inflammation signature[56]; NES, normalized enrichment score. *n* = 3 biological replicates for the data in all panels. **a**, **c**, **e**, **h**, **i** Statistical test used: Wald test, using Benjamini-Hochberg test for adjustment for multiple comparisons. **b**, **d**, **f** Fisher's exact test. Reg: regulation, Resp: response, Pos: positive, Inflam: inflammatory, Neg: negative, Prod: production, Ext: external, Bact: bacterial, Def: defense, NO: nitric oxide, Stim: stimulation.

correlated GO terms (Fig. 1g), demonstrating a deficient capacity of p30 to specifically activate inflammatory genes compared to p42-expressing cells. Transcription of inflammatory genes is mainly controlled by three TF families: NF-κB, interferon, and AP-1. As CEBPA has been shown to control upstream expression of TFs[48,49], we wondered whether CEBPA isoforms were differentially controlling expression of inflammatory TFs. Interestingly, the fraction of up- and down-regulation of inflammatory TFs was different for the three families (Fig. 1h), but the strongest and most consistent change observed was the downregulation of the AP-1 family of TFs in p30-expressing cells (Fig. 1i).

Altered inflammatory gene expression is emerging as a key pathogenic factor in pre-leukemic myeloid malignancies and AML[43,50,51]. Although both the origin and the consequences of these alterations are incompletely understood, they seem to be mutation- and age-specific[52–55]. A recent study has defined the set of inflammatory genes specifically deregulated in malignant cells from adult AML patients[56]. We sought to determine whether this adult-specific AML signature was dependent on p42 expression. For this, we performed

GSEA analysis, which showed a strongly downregulated expression of the AML-inflammation signature in p30-expressing HPC-7 cells and hematopoietic progenitors (NES = -2.1 and NES = -2.4, respectively, Fig. 1j, Supplementary Fig. 1q). This suggests that *CEBPA* mutations may interfere with inflammatory upregulation in human AML and may help explain the high heterogeneity of inflammatory expression among AML patients[56].

## p30 is associated with decreased inflammatory gene expression in AML

We next examined the transcriptomic profile of *CEBPA* mutants in AML. We first compared gene expression profiles of 50 patients with *CEBPA* mutations versus 758 patients without *CEBPA* mutations[22,23]. GSEA confirmed a significant downregulation of inflammatory genes in *CEBPA* mutant AML (NES = -1.95, FDR = 0.003; Fig. 2a), and down-regulated genes significantly overlapped those in p30-expressing HPC-7 cells (p-Val = 1.39e-15, Supplementary Fig. 2a). We then classified *CEBPA*-mutant patients into C-terminal, N-terminal, and biallelic N/C-terminal mutants and performed differential expression analysis

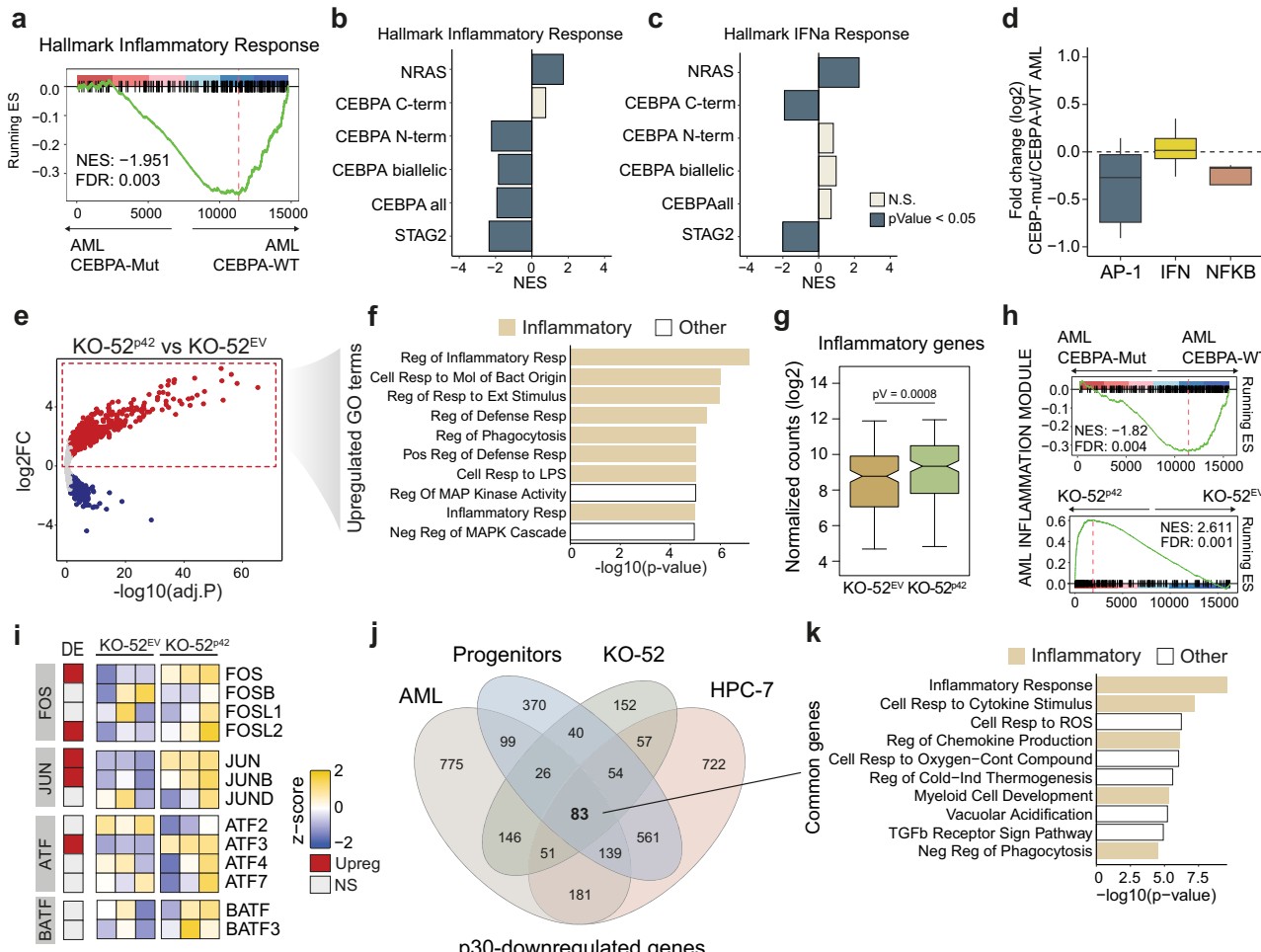

**Fig. 2 | Reduced inflammatory gene expression in *CEBPA*-mutant AML. a** GSEA of inflammatory response genes in *CEBPA*-mutant AML vs all the other AML mutants. Enrichment of (**b**) inflammatory response genes and (**c**) interferon alpha response genes in *NRAS*-, *CEBPA^NT^*-, *CEBPA^CT^*-, *CEBPA^bi^*- and *STAG2*- mutated AML, statistically tested by GSEA (NES=Normalized Enrichment Score). **d** Fold-change (log2) in expression of AP1/IFN/NFκB regulators in *CEBPA*-mutant AML. **e** Volcano plot of deregulated genes in KO-52^p42^ vs KO-52^EV^. Red: significantly upregulated genes (adj. p value < 0.05, log2FC > 1), blue: significantly downregulated genes (adj. p value < 0.05, log2FC < -1; *n* = 3 biological replicates). **f** Top 10 gene ontology terms of upregulated genes in KO-52^p42^ cells. **g** Normalized counts (rlog) of differentially expressed inflammatory genes in KO-52^EV^ and KO-52^p42^, *n* = 3 biological replicates, two-sided unpaired *t* test. **h** GSEA of an AML inflammation signature[56] in *CEBPA^bi/NT^*-mutated AML vs all the other AML mutants (top) and KO-52^p42^ vs KO-52^EV^ (bottom). NES normalized enrichment score. **i** Expression (z-score) of AP-1 family members

grouped by subfamily in three replicates (each individual column is a replicate) of KO-52^EV^ and KO-52^p42^. Left column (DE) indicates the differential expression status. Red: upregulated genes, blue: downregulated genes, gray: non-significantly changed genes (p-adj <0.05 log2FC <|0|). **j** Overlap of downregulated genes in p30-expressing cells in the different models. **k** Top 10 gene ontology terms enriched in p30-dependent genes common to: *CEBPA*-mutant AML, p30 hematopoietic progenitors, KO-52^p42^ and p30-HPC-7. *n* = 3 biological replicates. **d, g** Boxplots: center lines represent the median, lower and upper hinges correspond to the first and third quartiles (25th and 75th percentiles). Upper whisker extends from the hinge to the largest value no further than 1.5 * IQR from the hinge (IQR, inter-quartile range). Lower whisker extends from the hinge to the smallest value at most 1.5 * IQR of the hinge. **d, e, i** Statistical test used: Wald test, using Benjamini-Hochberg test for adjustment for multiple comparisons. **f, k** Fisher's exact test.

compared to all other AML cases. As controls of AML mutants known to alter inflammatory gene expression, we chose *NRAS* as an activating mutation[57,58] and *STAG2* as an inactivating mutation[59]. Inflammatory response genes were significantly downregulated in p30-expressing mutants (N-terminal and biallelic) but not in C-terminal mutants, which predominantly express p42 (Fig. 2b). In contrast, the interferon pathway was downregulated in C-terminal mutants but not in p30-expressing mutants, indicating distinct effects within the different inflammatory pathways (Fig. 2c). Consistent with this, we observed a marked downregulation of AP-1 TFs, a milder downregulation of NF-κB TFs, and no clear tendency towards up- or downregulation of interferon TFs (Fig. 2d).

To assess whether the changes in inflammatory gene expression were a direct consequence of the absence of p42 in *CEBPA*-mutants, we transiently transfected wild-type p42 in KO-52 cells, which are AML

cells carrying biallelic *CEBPA* mutations[37]. A transient transfection allowed us to analyze gene expression changes shortly after the initial expression of the protein, preventing premature proliferation arrest or induced differentiation that could confound the results (Supplementary Fig. 2b, c). Using RNA spike-ins for normalization, we identified 2611 upregulated genes in p42-expressing KO-52 cells (Fig. 2e), which were distinctly enriched in inflammatory genes, confirming their rescue upon p42 reconstitution (Fig. 2f, g, Supplementary Fig. 2d, e). Similarly, the inflammatory AML module, which we found to be downregulated in *CEBPA^bi/NT^*-mutated AML, recovered expression after p42 transfection (NES = 2.61, FDR = 0.001, Fig. 2h). We then focused on AP-1 family members, which were the most downregulated inflammatory TFs in CEBPA mutant AML (Fig. 2d). Out of 13 analyzed members of the main AP-1 subfamilies (FOS, JUN, ATF and BATF), five of them (38%) were significantly upregulated upon p42 reintroduction

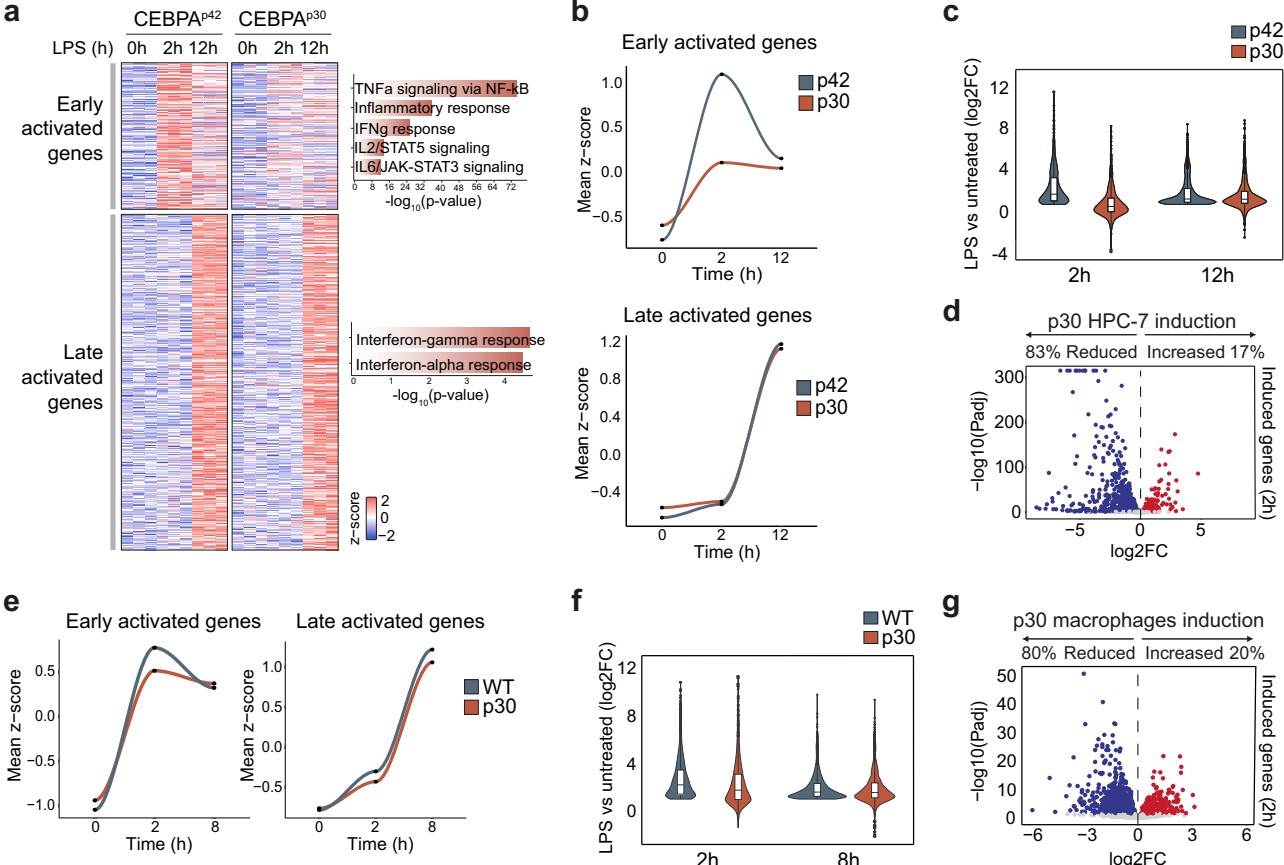

**Fig. 3 | Impaired LPS-activation dynamics in p30-expressing cells. a** Expression (z-score) of early and late activated genes defined by upregulation in p42 HPC-7 cells after 2 h (early) or 12 h (late) of LPS. Right: significantly enriched GO terms in early and late activated genes (Fisher's exact test). **b** Expression (mean z-score) of the early activated genes (top) and late activated genes (bottom) at the different LPS stimulation timepoints (0, 2 and 12 h) in p42 HPC-7 and p30 HPC-7. **c** LPS-induction (log2 fold change of LPS-treated cells versus untreated cells) of p42-upregulated genes at 2 h and 12 h of LPS stimulation in p42 and p30 HPC-7 cells. **d** Volcano plot of differential gene expression (p30 vs p42 at 2 h of LPS), showing only genes induced at 2 h in p42 HPC-7 cells. In red, upregulated genes and in blue, downregulated genes, p value < 0.05, log2FC <|0|. **e** Expression (mean z-score) of the early activated genes (left) and late activated genes (right) at the different LPS stimulation timepoints (0, 2 and 8 h) in *Cebpa^{Fl/Fl}* (WT) and *Cebpa^{Fl/p30}*; R26-CreER (p30) macrophages. **f** LPS-induction (log2 fold change of LPS-treated cells versus untreated cells) of WT-upregulated genes at 2 h and 8 h of LPS stimulation in *Cebpa^{Fl/Fl}* (WT) and *Cebpa^{Fl/p30}*; R26-CreER (p30) macrophages. **g** Volcano plot of differential gene expression (p30 vs WT at 2 h of LPS), showing only genes induced at 2 h in WT macrophages. In red, upregulated genes and in blue, downregulated genes, p value < 0.05, log2FC <|0|. *n* = 3 biological replicates. **c, f** Boxplots: center lines represent the median, lower and upper hinges correspond to the first and third quartiles (25th and 75th percentiles). Upper whisker extends from the hinge to the largest value no further than 1.5 * IQR from the hinge (IQR, inter-quartile range). Lower whisker extends from the hinge to the smallest value at most 1.5 * IQR of the hinge. **d, g** Statistical test used: Wald test, using Benjamini-Hochberg test for adjustment for multiple comparisons.

(adj.P < 0.05; Fig. 2i). Finally, we overlapped the p30-downregulated genes in human AML and mouse hematopoietic progenitors (Fig. 2j), and found that the 83 common genes to all systems were significantly enriched in inflammatory terms (Fig. 2k). These results remained consistent if the control cells were p30-overexpressing KO-52 cells instead of an empty vector control, confirming that p30 drives lower levels of inflammatory transcription compared to p42 (Supplementary Fig. 2b,f-i).

Overall, these data show that in human AML, *CEBPA* N-terminal and biallelic mutations - which result in the expression of p30 – are associated to a reduced expression of inflammatory genes, and this can be partially rescued by re-expression of the p42 wild-type isoform.

**Impaired activation of the early inflammatory response**
Given the observed reduction in baseline inflammatory levels of p30-expressing cells, we sought to determine the extent to which this would influence the response to the bacterial cell wall component lipopolysaccharide (LPS), a well-known trigger of inflammation that activates the Toll-like receptor 4 (TLR-4). RNA-seq analysis over a time

course of LPS exposure revealed a reduced response of p30-expressing HPC-7 cells to LPS at an early time-point (2 h), in which several signaling pathways are active, but not at a late time-point (12 h), where activation is mainly dependent on interferon[60] (Fig. 3a, b and Supplementary Fig. 3a). Most downregulated genes at baseline stayed downregulated after LPS, and the same was true for upregulated genes (Supplementary Fig. 3b). Accordingly, expression of multiple AP-1 members remained low throughout the time-course (Supplementary Fig. 3c). Gene inducibility was more markedly reduced at 2 h than at 12 h (Fig. 3c). Out of all the differentially induced genes at 2 h, 83% showed reduced induction and only 17% increased induction in p30-expressing cells (Fig. 3d).

We then assessed whether this pattern was consistent in other experimental systems. In primary macrophages, we also observed a significantly reduced activation of primary response genes (2 h) (Fig. 3e) and inducibility was more impaired at 2 h than at 8 h (Fig. 3f, g). We next examined the early LPS response in p30-expressing primary hematopoietic progenitors and in the AML cell line KO-52. In both cellular systems, the absence of a functional p42 resulted in a

diminished upregulation of early response genes and a lower degree of inducibility (Supplementary Fig. 3d–g), underscoring the role of p42 in promoting a transcriptional state that enables a correct inflammatory response.

In summary, these findings indicate that cells expressing the p42 isoform elicit a more robust early response to immune stimuli compared to cells expressing the shorter, AML-associated, p30 isoform.

## Reduced cytokine secretion by p30-expressing cells

We noticed that among the inflammatory genes affected by p30, many were genes encoding secreted proteins, such as cytokines and chemokines (Fig. 4a). We next asked if these changes were translated into changes in cytokine production and secretion upon inflammatory stimulation. To address this, we analyzed secreted proteins by liquid chromatography coupled to mass spectrometry (LC-MS/MS). Following 16 h of LPS stimulation of HPC-7 cells, most downregulated proteins were related to inflammatory and immune responses (Fig. 4b), and the top 5 terms depleted in p30 cells were immune/inflammatory (Supplementary Fig. 4a). Most proteins belonging to the 'cytokine' gene set were less abundant in the p30 than in the p42 supernatant, including chemokines (CXCL2, CXCL3, CCL2, CCL9), and interleukins (IL6, IL12b) (Fig. 4c). Time-course secretome analysis revealed that several of these proteins were quantified already at 8 hours in the p42 supernatant, but not in p30 supernatants (Fig. 4d). These results were confirmed by antibody arrays (Supplementary Fig. 4b).

## Reduced fitness of p42-expressing cells after prolonged LPS exposure

As the persistent activation of inflammatory pathways leads to HSC functional decline[43], we hypothesized that the incomplete activation of inflammatory genes in p30-expressing cells might reduce the adverse effects of prolonged LPS exposure. To address this, we plated primary bone marrow hematopoietic progenitors in 4-OHT-containing methylcellulose—to induce the deletion of the WT allele in p30 progenitors (Supplementary Fig. 4c)—and supplemented with LPS. As expected, wild-type cells showed a reduction in colony number in LPS-treated methylcellulose. In contrast, p30-expressing cells were unaffected by LPS and showed a similar number of colonies compared to the control condition (Fig. 4e). These differences affected colony number but not colony type, which mainly consisted of granulocyte-macrophage colony-forming units (CFU-GM). We next assessed cell growth of HPC-7 cells in liquid culture exposed to LPS for one week. Consistent with its capacity to repress the E2F complex[61], p42 induced a slower proliferation rate compared to p30 at baseline, but LPS treatment further reduced their growth. However, LPS exposure did not affect the growth curve of p30 cells (Fig. 4f-g, Supplementary Fig. 4d), overall indicating that p30 cell growth is less affected from long-term LPS exposure compared to p42 cells. To determine if these observations were consistent in an AML cellular model, we assessed the colony-forming capacity of KO-52 cells overexpressing either wild-type p42 or p30 (as a control) in methylcellulose in the presence of LPS. While LPS only minimally reduced the number of colonies formed by p30-expressing cells, it significantly reduced the number of colonies formed by p42-expressing cells (Supplementary Fig. 4e), which is consistent with the reduced fitness in mouse primary progenitors (Fig. 4e) and HPC-7 cells exposed to LPS (Fig. 4g).

To understand the cause of this differing effect on cellular fitness, we noticed that the gene expression signature 'Apoptosis' was downregulated in CEBPA*bi/NT*-mutants compared to the rest of AML patients (Supplementary Fig. 4f), which is consistent with apoptosis being one of the detrimental effects of persistent inflammation[44]. To assess the fraction of apoptotic cells in p42 and p30-expressing cells, cells were plated in LPS-containing media and apoptosis was quantified by annexin V staining. In hematopoietic progenitors the difference was not statistically significant, but on average WT progenitor cells died more than p30 cells (Fig. 4h). The difference was clearer in HPC-7 cells expressing p42, which showed increased annexin V signal while p30 annexin V signal did not change (Fig. 4i). Collectively, these results show that cells expressing p42 are more vulnerable to the deleterious effects of persistent inflammation, which may represent a selective advantage for *CEBPA* mutant clones in AML.

## Chromatin profiling reveals differential binding of AP-1 members

To elucidate the mechanisms underlying the distinct effects of p30 and p42 on the inflammatory response, we conducted ChIP-seq of p42 and p30 in HPC-7 cells. A majority of CEBPA binding sites were found in intergenic and intronic regulatory regions (Fig. 5a) and, consistent with a previous study[35], most binding sites were shared between the two isoforms (Supplementary Fig. 5a). We then assessed CEBPA binding on the promoters of inflammatory genes (Fig. 5b) or intergenic peaks near inflammatory genes (Fig. 5c). Both isoforms were found at 4–5% of inflammatory gene promoters and, strikingly, 40–50% of inflammatory genes had nearby CEBPA peaks. To validate these findings, we compared the ChIP-seq signal obtained in HPC-7 cells (using two independent antibodies) to published p30 and p42 ChIP-seq data in GMPs expressing endogenous proteins[35], showing a strong enrichment at those peaks (Supplementary Fig. 5b). Moreover, the genomic distribution around inflammatory genes in the GMP ChIP-seq was comparable to that in HPC-7 (Supplementary Fig. 5c-e). Overall, this supports a direct role for CEBPA in regulating inflammatory gene expression, but also indicates that both isoforms bind to the same fraction of inflammatory genes, and therefore binding does not explain the differing transcriptional levels.

We then explored how the binding of each isoform influenced gene expression. To identify the target genes distally regulated by each isoform, we performed H3K27 acetylation HiChIP in HPC-7 cells, which detects specific interactions between active regulatory elements and genes[62]. In general, we did not find global differences in number, strength or distance of chromatin interactions when comparing HiChIP contacts in p30 and p42-expressing cells. However, genes with stronger interactions in p30 cells also exhibited increased transcription, and viceversa (Fig. 5d). In addition, HiChIP performed on LPS-activated cells revealed that interactions stronger in p42 cells compared to p30 cells were enriched for genes related to the inflammatory response (Supplementary Fig. 5f). We then assessed expression of inflammatory genes with direct CEBPA binding at the promoter or distal binding with a significant HiChIP interaction. Out of 513 expressed inflammatory genes, 267 (52%) showed binding of CEBPA at their promoter and/or enhancer. Of those, 43 (16%) had exclusive binding of p42, 46 (17%) of p30, but the majority (178) had binding of both (66%). The genes bound only by p42 had a tendency to be more expressed in p42-expressing cells than in p30 cells, in contrast to the genes only bound by p30 which did not show a clear trend (Fig. 5e). Surprisingly, inflammatory genes bound by both isoforms were predominantly more expressed in p42-expressing cells compared to p30-expressing cells (54% of genes in this category), while only 19% were more expressed in p30-expressing cells, and the rest did not show significant changes (adj.P < 0.05, Fig. 5e). This difference was not due to weaker binding by p30, as the ChIP-seq signal for p30 was slightly stronger at these sites (Supplementary Fig. 5g). An example of this is the LPS-responsive *Cxcl10* gene, which is located within a topologically associating domain (TAD) that contains other genes of the same family of inflammatory cytokines. *Cxcl10* is more strongly induced by LPS in p42-expressing compared to p30-expressing cells, despite both isoforms binding very similarly to regulatory sites in this region (Fig. 5f). Moreover, we observed interactions formed in p42-expressing cells anchored at p42 binding sites, whereas the same sites bound by p30 were unable to establish such interactions (Fig. 5f). Collectively, these results demonstrate that although both isoforms can bind to the same

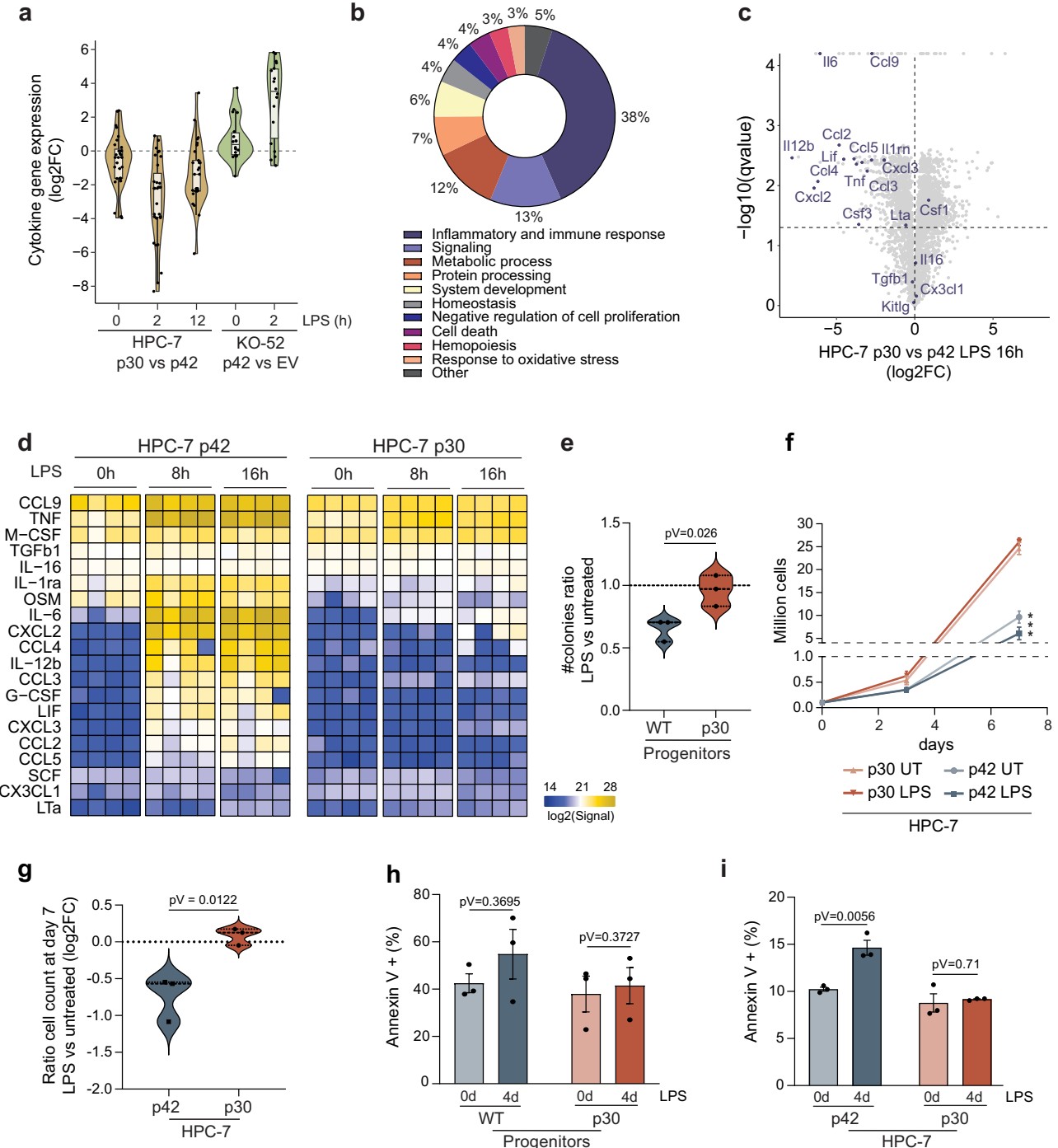

**Fig. 4 | Altered inflammatory phenotype of p30-expressing cells. a** Fold change of expression of cytokine-expressing genes in p30 HPC-7 vs p42 HPC-7 and KO-52[p42] vs KO-52[EV] during LPS treatment. Three biological replicates per timepoint were performed; center line at median, lower and upper hinges correspond to the first and third quartiles (25th and 75th percentiles). Upper whisker extends from the hinge to the largest value no further than 1.5 * IQR from the hinge (IQR, inter-quartile range). Lower whisker extends from the hinge to the smallest value at most 1.5 * IQR of the hinge. **b** Downregulated terms by GSEA (p30 vs p42, NES < 0, FDR < 0.05) of secreted proteins detected by mass spectrometry in supernatants of HPC-7 cells treated with LPS for 16 h. **c** Volcano plot of deregulated secreted proteins detected by mass spectrometry in p30 vs p42 supernatants treated with LPS for 16 h, highlighting cytokines. Unpaired two-sided Welch's T test to generate p values and permutation based FDR, 250 randomizations. **d** Cytokine secretion levels detected by mass spectrometry over an LPS time course of 0, 8 and 16 hours, each column is a replica. **e** Ratio of colony number in LPS vs untreated conditions in WT and p30 hematopoietic progenitors. **f** HPC-7 cell count in the presence of 100 ng/mL LPS. Two-sided paired t test, ***p value = 0.0005. **g** Ratio of cell number of LPS vs untreated conditions in p42 and p30 HPC-7 cells at day 7. Violin plots show median at center and quartiles. **h** Percentage of total annexin V+ cells in WT and p30 hematopoietic progenitors after 4 days of LPS stimulation. **i** Percentage of total annexin V+ cells in p42 and p30 HPC-7 cells after 4 days of LPS stimulation. **e–i** n = 3 biological replicates, data represented as mean +/− SEM. For statistical significance, two-sided t tests were used, unpaired (**e**, **g**, **i**) or paired (**h**). Source data are provided as a Source Data file.

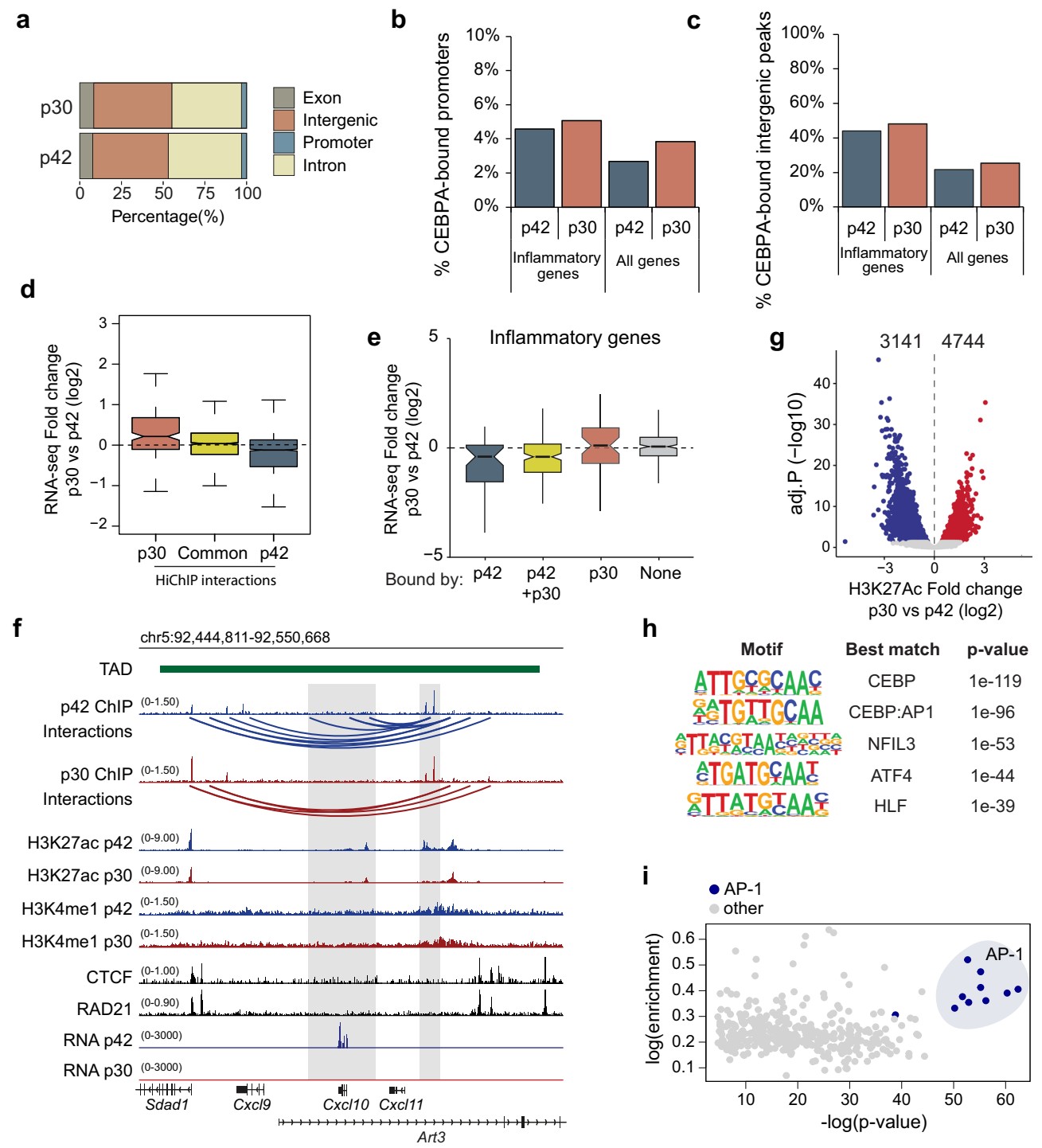

**Fig. 5 | Enhancer activation analysis reveals reduced AP-1 signature in p30-expressing cells. a** Genomic distribution of p30 and p42 ChIP-seq peaks in HPC-7 cells. **b** Percentage of genes with CEBPA binding in their promoters in HPC-7. **c** Percentage of genes most proximal to enhancers with CEBPA binding in HPC-7; enhancers were defined as peaks that are not in promoters and have a H3K27ac peak. **d** Differential gene expression (log2 fold change, p30 vs p42) of genes regulated by p42-specific promoter-enhancer interactions, common (p42 + p30) interactions, and p30-specific interactions. *n* = 3 (RNA-seq) and *n* = 2 (HiChIP) **e** Differential expression (log2 fold change, p30 vs p42) of inflammatory genes bound by p42 at their promoter and/or enhancer; bound by p42 and p30 in promoter/enhancer; bound by p30 at their promoter/enhancer; or of genes not bound by CEBPA. RNA-seq *n* = 3 and CEBPA ChIP-seq *n* = 1. **d, e** Boxplots: median at center line, lower and upper hinges are the first and third quartiles (25th and 75th percentiles). Upper whisker extends from the hinge to the largest value no further than 1.5*IQR from the hinge (IQR, inter-quartile range). Lower whisker

extends from the hinge to the smallest value at most 1.5*IQR of the hinge. **f** Browser view of the *Cxcl10* locus and its topologically associating domain (TAD), delimited by binding of CTCF and RAD21 (black tracks) in HPC-7 cells[47]; in blue, tracks from p42-expressing cells; in red, from p30-expressing cells. Top to bottom: ChIP-seq defining CEBPA binding sites, H3K27ac HiChIP loops, H3K27ac ChIP-seq, H3K4me1 ChIP-seq, CTCF and RAD21 ChIP-seq and RNA-seq of LPS-treated HPC-7 cells. Shaded in gray, *Cxcl10* promoter and enhancer (right) regions, with binding of both p42 and p30 at the enhancer. **g** Differential H3K27ac peaks in HPC-7 cells. In red upregulated genes; in blue downregulated genes; in gray non-significantly changed genes. p-adj<0.05, log2FC <|0|, Wald test, Benjamini-Hochberg adjustment for multiple comparisons. **h** Top 5 enriched transcription factor motifs among downregulated H3K27ac peaks (excluding promoters) in p30 untreated cells. **i** Enriched TF motifs in inflammatory enhancers that fail to be induced in p30 cells. **h, i** Enrichment is calculated using the cumulative binomial distribution (Homer).

sites, they exert distinct effects on gene expression, especially within the inflammatory gene set.

To understand the factors dictating the differences in regulatory behavior between p30 and p42, we explored whether additional elements were contributing to the differential activation of inflammatory genes. To this end, we performed differential binding analysis of H3K27ac ChIP-seq signal comparing p42 and p30-expressing cells, revealing 4744 increased peaks and 3141 reduced H3K27ac peaks in p30 cells (Fig. 5g). Motif enrichment analysis among downregulated enhancers revealed that the most significantly enriched motif was the CEBP motif (Fig. 5h), consistent with the observation that a fraction of sites commonly bound by the two isoforms was more activated by p42 than by p30. Interestingly, the second most significantly enriched motif was the composite CEBP:AP1 motif, which is bound by dimers of both TF families (Fig. 5h). Next, we focused on inflammatory enhancers. For this, we first identified enhancers induced by LPS in p42 cells and classified them into those that are also induced by LPS in p30 cells and those that fail to be induced in p30 cells. Strikingly, motif

enrichment analysis among the p30-failed inflammatory enhancers showed that among the top ten enriched motifs, nine were AP-1 protein motifs (Fig. 5i). Overall, these data showed that the altered enhancer activation detected upon binding of p30 around inflammatory genes may be due to impaired AP-1 binding.

## FOS is downregulated in mutant CEBPA cells and is partially responsible for the decreased inflammatory expression

AP-1 is an extensive family of transcription factors with essential roles in stress response, signal transduction and inflammation[63,64]. Since the AP-1 binding motif was significantly associated to enhancers downregulated in p30 cells, we examined expression of the main four AP-1 subfamilies in p30-expressing HPC-7 cells, macrophages, hematopoietic progenitors and $CEBPA^{NT/bi}$-mutant AML, and we observed that the FOS subfamily was consistently the most downregulated by p30 (Supplementary Fig. 6a). Across the four systems, the FOS TF and its paralogs FOSL1, FOSL2 and FOSB showed decreased levels of expression in p30-expressing cells (Fig. 6a). AML patients with *CEBPA*

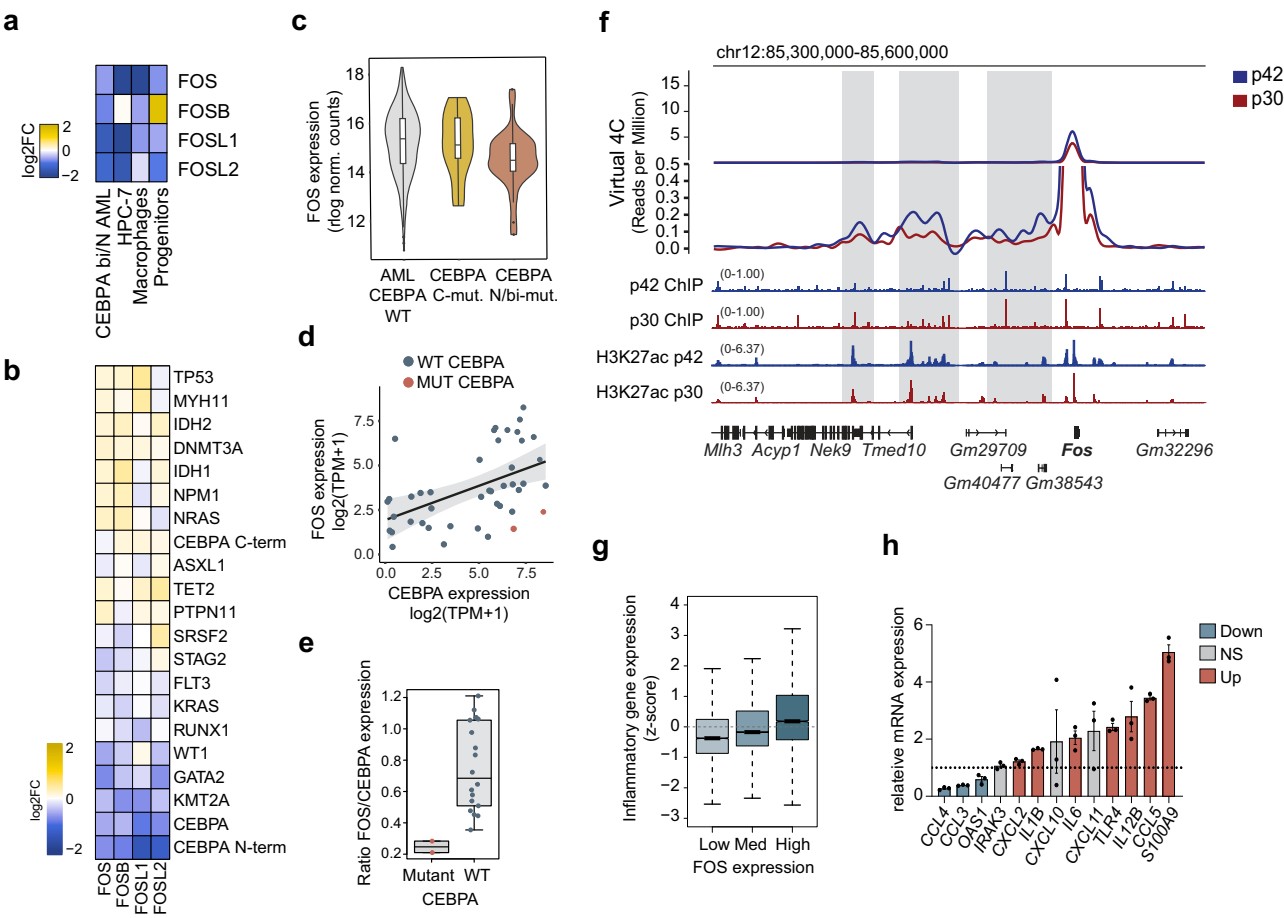

**Fig. 6 | *FOS* transcription is differentially regulated by p30/p42 and partially explains the reduced inflammatory expression. a** Differential gene expression of FOS subfamily members in various experimental models: *CEBPA*[bi/NT]-mutated AML compared to all the other AML subtypes, p30 vs p42 HPC-7 cells, *Cebpa*[Fl/p30];R26-CreER (p30) macrophages versus *Cebpa*[Fl/Fl] (WT) macrophages, and in *Cebpa*[Fl/p30];R26-CreER (p30) vs *Cebpa*[Fl/Fl] (WT) bone marrow (BM) hematopoietic progenitors. **b** Log2 fold-change of gene expression of FOS subfamily members in each AML mutant compared to the rest of AML patients WT for that particular mutation. **c** *FOS* expression (normalized counts at log2 scale) in AML patients with wild-type CEBPA (*n* = 759), CEBPA C-terminal mutations (*n* = 17) or biallelic and N-terminal CEBPA mutations (*n* = 32). **d** Correlation of *FOS* and *CEBPA* gene expression in AML cell lines from DepMap[65]. **e** Ratio of *FOS*/*CEBPA* gene expression in *CEBPA*-mutant (*n* = 2) and wild-type (*n* = 18) AML cell lines. Linear regression with standard error. **f** Virtual 4 C of the *Fos* gene locus in p42 and p30-expressing HPC-7

cells. Tracks: ChiP-seq of p42 (blue) and p30 (red); ChIP-seq of H3K27ac in p42 (blue) and p30 (red) expressing cells. **g** Expression (z-score) of inflammatory genes in AML patients stratified by *FOS* expression. Low and high represent the bottom and top 50 patients by *FOS* expression, respectively. **h** Quantitative RT-PCR of representative inflammatory genes in KO-52 cells transduced with FOS, expression relative to an empty vector control (dotted line) and normalized to *HPRT* and *RPL38*. In blue, downregulated genes; in red, upregulated (p value < 0.05) and in gray, non-significant changes (p value > 0.05), two-tailed unpaired t test; mean ± SEM, *n* = 3 biological replicates. **c**, **e**, **g** Boxplots center lines represent the median, lower and upper hinges correspond to the first and third quartiles (25th and 75th percentiles). Upper whisker extends from the hinge to the largest value no further than 1.5 * IQR from the hinge (IQR, inter-quartile range). Lower whisker extends from the hinge to the smallest value at most 1.5 * IQR of the hinge. Source data are provided as a Source Data file.

mutations show the lowest expression levels of FOS family genes of all AML types grouped by mutation (Fig. 6b). Consistent with the data shown throughout the study, biallelic and N-terminal but not C-terminal CEBPA mutations are associated to decreased *FOS* expression (Fig. 6b, c). Although there is a positive correlation between *CEBPA* and *FOS* expression in AML cell lines present in DepMap[65] (Fig. 6d), the only two *CEBPA*-mutant cell lines (KO-52 and Kasumi-6) show the lowest ratio of FOS/CEBPA expression of the top 20 highest-expressing CEBPA cell lines (Fig. 6d, e). Overall, these data indicate that mutations in CEBPA are associated with decreased FOS expression in AML.

To understand how p42 and p30 differentially regulate FOS expression, we examined the *Fos* locus. While both proteins bind to the same enhancers, p42 elicits a stronger H3K27ac signal, inducing a stronger interaction with the gene promoter as shown by virtual 4 C from HiChIP signal (Fig. 6f). To determine whether this holds true in AML, we reanalysed Hi-C data[66] from AML patients with *CEBPA* mutations (N = 6) and compared them with AML patients lacking *CEBPA* mutations (N = 16). Virtual 4C-seq around the human *FOS* locus confirmed reduced interactions with upstream regulatory regions in *CEBPA*-mutant AML (Supplementary Fig. 6b). We then hypothesized that rescuing p42 expression in *CEBPA*-mutant AML cells should increase FOS expression. Indeed, KO-52 cells that express wild-type p42 compared to mock-transfected KO-52 as well as compared to p30-transfected control cells showed a strong upregulation of *FOS* gene expression (Fig. 2i and Supplementary Fig. 2i), confirming that p42 drives higher *FOS* expression levels than p30. To gain a deeper molecular understanding of this rescue, we performed ATAC-seq in p42- and p30-transduced KO-52 cells, identifying 11,674 peaks with increased accessibility in p42 cells (adj. P < 0.05) (Supplementary Fig. 6c). Motif enrichment analysis of these peaks closely resembled the enriched motifs among p42-induced H3K27ac peaks in HPC-7 cells shown in Fig. 5h, with the CEBP:AP1 motif among the top five most significantly enriched (Supplementary Fig. 6d). To determine whether FOS was one of the AP-1 factors binding these sites, we plotted the fold-change in accessibility across all sites, revealing that FOS-bound sites were associated with higher ATAC signal (Supplementary Fig. 6e). Next, we identified genes displaying increased promoter accessibility, separating those with (N = 154) and without (N = 394) FOS binding. Accordingly, the genes bound by FOS were more strongly upregulated in p42 cells (Supplementary Fig. 6f). GO analysis showed that FOS-bound genes were enriched for inflammatory terms, suggesting a role for FOS in enhancing accessibility at inflammatory promoters (Supplementary Fig. 6g). Finally, regardless of accessibility, inflammatory genes co-bound by CEBPA and FOS exhibited a greater expression increase in p42-transfected cells compared to p30-transfected cells than did CEBPA-FOS co-bound house-keeping genes[67] lacking an inflammatory function or inflammatory genes not bound by these factors, with an even stronger effect after LPS stimulation (Supplementary Fig. 6h). All these data suggest that FOS upregulation in p42-expressing cells drives increased chromatin accessibility and expression of inflammatory genes, providing a mechanistic basis for the partial rescue of inflammatory transcription in p42 cells.

In line with the known role of AP-1 proteins in the inflammatory response[64,68], AML patients with high *FOS* expression display higher levels of inflammatory gene expression than patients with low *FOS* expression (Fig. 6g). Therefore, we asked whether the lower expression of FOS in p30-expressing cells may underlie the altered inflammatory response. To address this, we examined the effect of FOS overexpression (Supplementary Fig. 6i) on the transcription of a panel of representative inflammatory genes. Forcing expression of FOS induced a significant upregulation of ~50% of the tested inflammatory genes (p.val <0.05), including well-known inflammatory mediators such as *CCL5* and *IL6* (Fig. 6h). A similar trend was observed in LPS-stimulated cells (Supplementary Fig. 6j), overall implicating FOS as

an intermediary of p42-mediated inflammatory gene expression. We conclude that FOS downregulation partially explains the reduced inflammatory response in p30-expressing cells.

To assess the contribution of FOS to cellular fitness in response to inflammatory signals, we plated FOS-overexpressing KO-52 cells in LPS-supplemented methylcellulose. LPS treatment caused a more pronounced reduction in the number of colonies formed by FOS-expressing cells compared to control cells (Supplementary Fig. 6k,l). Since we noticed that LPS reduced both the colony number and the total cells in the initial plating, we calculated the cumulative CFU potential, which accounts for the initial expansion difference in the overall clonogenic activity[69]. This showed that while control cells showed minimal reduction when exposed to LPS, FOS-expressing cells presented a significantly reduced CFU potential (Supplementary Fig. 6m,n). Overall, these experiments indicate that the partial rescue of inflammatory gene expression by FOS leads to a greater fitness reduction in FOS-expressing cells under inflammatory conditions.

## ATF4 preferentially interacts with p42

Exploring additional AP-1 factors, we noticed a significant enrichment of the ATF4 motif among enhancers that failed to be activated in p30-expressing HPC-7 cells. It was the fourth most significantly enriched motif in untreated cells, but it was the most significantly enriched motif after the CEBP motif when we filtered for enhancers exclusively downregulated after LPS stimulation (Fig. 7a). ATF4 is an AP-1 transcription factor that plays a pivotal role in the activation of the integrated stress response (ISR), which besides viral infections can also be triggered by endoplasmic reticulum (ER) stress, among others[70]. In the hematopoietic system, ATF4 is particularly important for HSC homeostasis, as the response to stress is essential for maintaining clonal integrity and preventing the propagation of damaged stem cells[71,72]. We first examined the H3K27ac signal around ATF4 ChIP-seq binding sites and, as expected, p42-expressing cells exhibited higher acetylation levels (Fig. 7b, c). We next assessed protein abundance by WB but only detected a slight but not significant reduction in ATF4 protein levels in p30 cells (Supplementary Fig. 7a). As ATF4 can dimerize with CEBP proteins[73,74], we asked whether it would dimerize with p42 and p30. To address this, we immunoprecipitated CEBPA in cells overexpressing either FLAG-tagged p42 or p30. Western blot of ATF4 showed an interaction with p42 but not with p30, indicating the inability of p30 to interact with ATF4 (Fig. 7d). To gain deeper mechanistic insights, we next asked whether the biochemical properties of CEBPA's N-terminal region influence its interaction with ATF4. This region contains a large intrinsically disordered region (IDR) that undergoes phase separation[75]. We expressed a mutant version in which all aromatic residues in the IDR are mutated, strongly impairing the phase separation capacity (AroLite)[75]. Co-immunoprecipitation showed markedly reduced ATF4 signal in the AroLite IP compared to wild-type CEBPA (Supplementary Fig. 7b). Conversely, a mutant with perfectly spaced aromatic residues that shows increased phase separation[76] (AroPerfect) exhibited enhanced ATF4 binding (Supplementary Fig. 7b). Moreover, deleting the only transactivation domain (TE-III) that is shared by p30 and p42[77] also decreased the interaction with ATF4. Together, these results indicate that the sequence and properties of the N-terminal region are required for ATF4 interaction, and suggest a potential role of phase separation in the formation of this complex.

To assess if p30-expressing cells fail to properly activate ATF4-target genes, we examined genes with ATF4 binding either at the promoter or at distal regulatory regions bound by a HiChIP loop. We found that these genes were in general more expressed in p42 than in p30 compared to the rest of genes not bound by ATF4 (Fig. 7e). Among those, genes bound by both ATF4 and CEBPA were the main drivers of this expression change, compared to genes bound by either factor

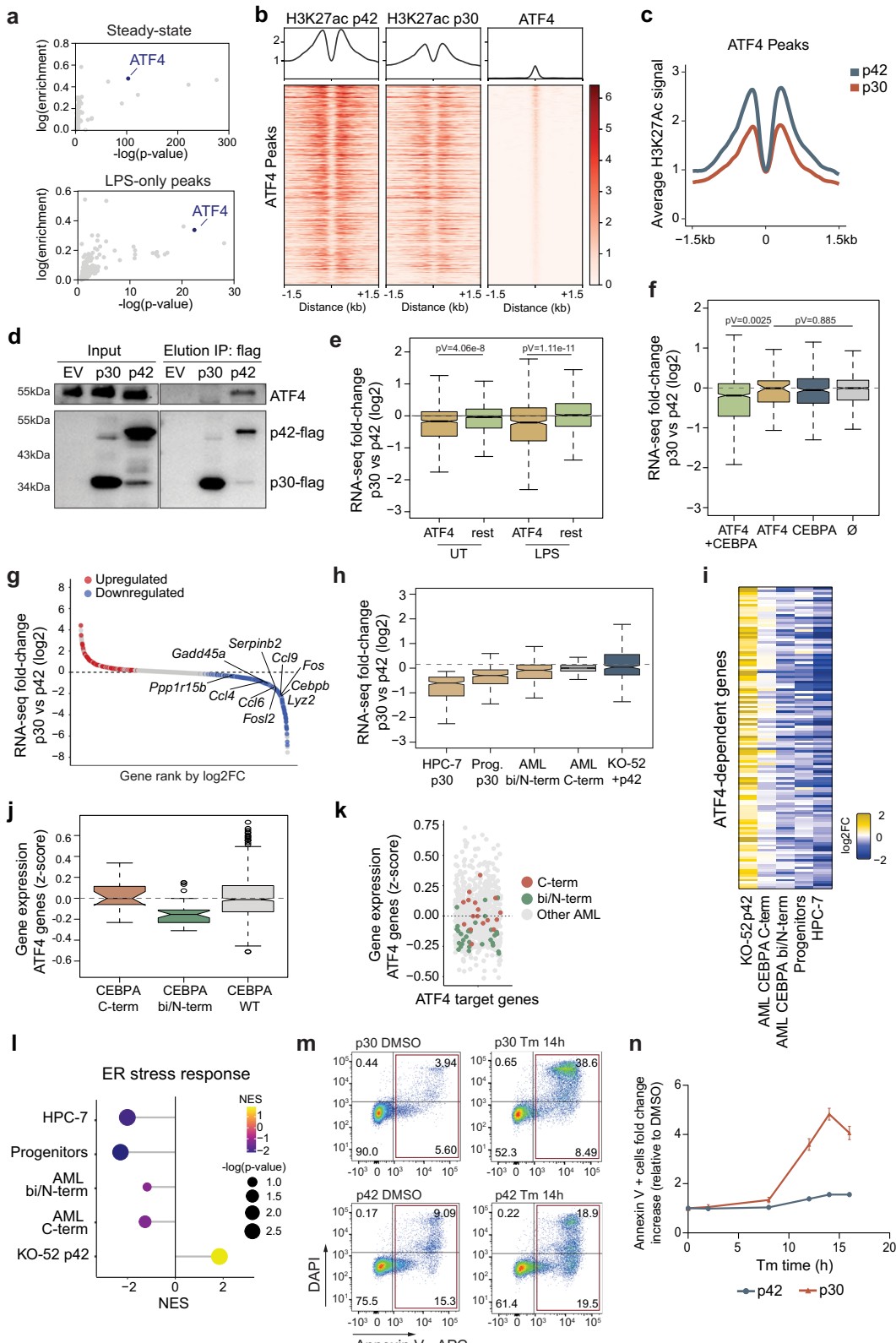

alone (Fig. 7f). The 368 CEBPA-ATF4 co-regulated genes showed a general trend of being more expressed in p42 than in p30, although other genes (121) showed the opposite behavior, consistent with a complex behavior of ATF4 in gene expression regulation[78] (Fig. 7g). Interestingly, the downregulated group was enriched in inflammatory response terms (Supplementary Fig. 7c), and included key inflammatory genes such as *Fos*, *Cebpb* and *Ccl9* as well as ISR effector genes like

*Gadd45a* and *Ppp1r15b* (Fig. 7g). Genes in this group were also mostly downregulated in hematopoietic progenitors and CEBPA-biallelic and N-terminal AML but not in C-terminal AML and KO52-p42, where they were mostly upregulated (Fig. 7h, i). In order to further determine the involvement of ATF4 in the regulation of p42-dependent genes, we used shRNAs to knock down *Atf4* in HPC-7 cells expressing either p42 or p30 (Supplementary Fig. 7d). We then analysed gene expression by

**Fig. 7 | ATF4 associates with p42 but not p30. a** Motif enrichment (cumulative binomial distribution) among H3K27ac downregulated peaks in p30 HPC-7 in untreated cells (steady-state) and H3K27ac peaks exclusively downregulated in LPS-treated samples. Peaks overlapping promoters were excluded. **b** H3K27ac ChIP-seq signal around ATF4 peaks in p42 (left) and p30-expressing HPC-7cells (middle), ATF4 ChIP-seq signal is plotted in the right. **c** Average H3K27ac ChIP-seq signal at ATF4 peaks in p42- and p30-expressing cells. **d** Immuno-blot against CEBPA-Flag and ATF4 of the immunoprecipitation of CEBPA-Flag in HEK293F cells over-expressing an empty vector (EV), p30 or p42. **e** Differential gene expression (log2 fold change, p30 vs p42 HPC-7) of genes bound by ATF4 at distal regulatory regions bound by a HiChIP loop, and the rest of genes not bound by ATF4 in untreated and stimulated with LPS for 2 h cells. **f** Differential gene expression (log2 fold change, p30 vs p42) of genes connected by a HiChIP-defined loop to peaks of the combination of p42, p30 (CEBPA) and ATF4; genes bound only by ATF4; genes only bound by CEBPA and of genes bound by none. **e**, **f** p values calculated with two-sided unpaired t test. **g** Genes connected by a HiChIP-defined loop to peaks of CEBPA and ATF4 ranked by fold-change of p30 vs p42. **h** Differential expression (log2 fold change) of the downregulated CEBPA-ATF4 co-bound genes (g) in p30 vs p42 HPC-7, *Cebpa*$^{Fl/p30}$;R26-CreER (p30) vs *Cebpa*$^{Fl/Fl}$ (WT) progenitors, *CEBPA*$^{bi/NT}$- and *CEBPA*$^{CT}$-mutated AML compared to all other AML subtypes, and in KO-52$^{p42}$ vs

KO-52$^{EV}$. **i** Heatmap of the expression of CEBPA-ATF4 co-dependent genes (g) that are downregulated in p30 HPC-7 and CEBPA$^{bi/NT}$ AML and upregulated in KO-52$^{p42}$, using the same groups as in (**h**). **j** Mean expression of ATF4 target genes in *CEBPA*$^{CT}$-*CEBPA*$^{bi/NT}$- mutated AML (*n* = 17 and *n* = 32, respectively) and in AML patients with wild-type *CEBPA* (*n* = 759). **k** Distribution of AML patients based on ATF4 target mean gene expression. **l** Enrichment of 'Response to ER stress' gene set in p30 vs p42 HPC-7, *Cebpa*$^{Fl/p30}$;R26-CreER (p30) vs *Cebpa*$^{Fl/Fl}$ (WT) progenitors, *CEBPA*$^{bi/NT}$-mutated AML compared to all other AML subtypes and in KO-52$^{p42}$ vs KO-52$^{EV}$, statistically tested by GSEA (NES=Normalized Enrichment Score). **m** FACS plots of p30 and p42 HPC-7 stained with DAPI and annexin V-APC. Cells were treated with DMSO or 1 μg/mL of tunicamycin (Tm) for 14 h. **n** Fold change increase in the percentage of annexin V+ cells relative to DMSO over a time-course of Tm treatment, mean +/− SEM, *n* = 3 biological replicates. **e**, **f**, **h** Data from RNA-seq with *n* = 3 biological replicates, HiChIP performed in duplicate and publicly available ATF4 ChIP-seq; boxplots center line at median, lower and upper hinges correspond to first and third quartiles. Upper whisker extends from the hinge to the largest value no further than 1.5*IQR from the hinge (IQR, inter-quartile range). Lower whisker extends from the hinge to the smallest value at most 1.5*IQR of the hinge. Source data are provided.

qPCR of genes previously identified as downregulated in p30 HPC-7 cells (compared to p42) and co-bound by CEBPA and ATF4. Upon *Atf4* knockdown, the differences between p42 and p30 became less pronounced or were even reversed (Supplementary Fig. 7e), supporting a specific cooperation between p42 and ATF4 in stress-response gene regulation. Notably, *Fos* is among the co-regulated genes (Fig. 7g, Supplementary Fig. 7e), linking the p42–ATF4 interaction to the reduced *Fos* levels and subsequent inflammatory gene deregulation observed in p30-expressing cells.

In AML, CEBPA biallelic mutants have reduced expression of ATF4 target genes compared to the rest of AML mutants (Fig. 7j, k). Indeed, the 50 bottom ATF4-target expressing patients (of a total of 808 AML patients) are significantly enriched for CEBPA biallelic mutations (pVal = 0.0002, binomial test). Collectively, these data indicate that p42 and ATF4 cooperate in regulating a set of genes—many of which are inflammatory—and that the inability of p30 and ATF4 to interact results in the deregulation of many of those genes.

Consistent with the essential role of ATF4 in the unfolded protein response, ER stress-response genes were down-regulated in p30-expressing cells (Fig. 7l). We then wondered whether cells lacking a functional p42-ATF4 dimer would be more susceptible to ER-stress-induced apoptosis. To this end, we treated HPC-7 cells with tunicamycin (Tm) to induce an acute ER stress response. Strikingly, peaking at 14 h of Tm treatment, total cell death in p42 cells increased by 1.5-fold compared to untreated cells, whereas it increased by almost five fold in p30-expressing cells (Fig. 7m, n). We next assessed the viability of KO-52 cells overexpressing either p42 or p30 in response to tunicamycin. A 5-day tunicamycin treatment triggered cell death in both p30- and p42-overexpressing KO-52 cells, but it was significantly more pronounced in p30-expressing cells, indicating a protective effect of p42 expression under ER stress (Supplementary Fig. 7f). We then examined the impact of ER stress on the self-renewal capacity of KO-52 cells in methylcellulose culture. Consistent with the cell survival analysis, p42-expressing cells formed significantly more colonies than p30-expressing cells (Supplementary Fig. 7g). Collectively, these results indicate that the acute induction of ATF4-regulated pathways can selectively induce cell death of p30-expressing cells, and suggest that this could be exploited as a vulnerability to treat leukemia patients with N-terminal *CEBPA* mutations.

Next, we performed ChIP-seq to identify ATF4 binding sites in KO-52 cells, yielding 16,470 peaks, and found that genes bound by ATF4 at their promoter or gene body were upregulated in p42-rescued KO-52 cells but not in p30-expressing cells (Supplementary Fig. 7h). We then conducted ATAC-seq on p30- and p42-expressing KO-52 cells under

steady-state conditions and after 8 h of tunicamycin treatment. At baseline, ATF4 binding sites were more accessible in p42-expressing cells compared to p30-expressing cells (Supplementary Fig. 7i). Following tunicamycin treatment, chromatin accessibility at ATF4 sites increased in both cell lines, but p42-expressing cells maintained higher accessibility than p30-expressing cells (Supplementary Fig. 7i). We then identified ATAC-seq peaks that either increased or decreased following tunicamycin treatment in p42-expressing KO-52. In both cases, the magnitude of these changes was greater in p42-expressing cells than in p30-expressing cells, where the stress response triggered milder changes, albeit in the same direction (Supplementary Fig. 7j). Together, these results suggest that p42 expression enhances accessibility at ATF4 binding sites and may drive differential expression of ATF4-target genes in *CEBPA*-mutant AML.

## Discussion

The predominant expression of the p30 isoform is observed in most CEBPA mutant AML cases. However, the regulatory properties of p30 remain incompletely understood. While it retains the DNA binding domain and the capacity to regulate gene expression, the transcriptional output of p30 activity is markedly different from that of p42. This is particularly surprising given that both proteins share most of their binding sites on chromatin[35]. A major obstacle to compare the expression of genes regulated by each isoform is that sorted primary cell populations may differ in terms of composition and differentiation stage, complicating the interpretation of the gene expression results. To overcome this limitation and comprehensively characterize the regulatory behavior of p30 and p42, we expressed them in an isogenic hematopoietic stem cell line (HPC-7 cells) in an inducible manner, ensuring that the epigenetic and transcriptomic background remained as similar as possible. To confirm that the results in HPC-7 cells were not restricted to this cell line, we also used primary cell populations as a reference as well as the rescued *CEBPA*-mutant AML cell line KO-52 and AML patient's data. Specifically, ex vivo differentiated macrophages allowed us to ensure that a potentially altered differentiation stage of progenitor populations was not a confounding factor.

Here we identify inflammation as a major transcriptional program differentially affected by mutant *CEBPA*. Apart from baseline inflammatory transcription, we demonstrate that the initial response to LPS is dampened. Baseline transcriptional changes can partly explain the deficient LPS activation, as AP-1 factors are downregulated before treatment while late response genes, such as interferon-regulated genes, remain largely unaffected. This effect is consistent across

**CEBPA isoform functions in stress responses**

Fig. 8 | **Proposed model of the interplay between CEBPA isoforms, AP-1 factors and external stimuli.** p30 is unable to promote normal expression of the *FOS* gene, leading to deregulation of FOS target genes, including many inflammatory genes. As a consequence, p30-expressing cells have reduced cytokine secretion and cell death compared to p42-expressing cells upon persistent inflammatory stress. We also observe that only the p42 isoform can dimerize with the AP-1 factor ATF4, crucial in the regulation of the ER stress response. Consequently, there is an altered transcriptional activation of ATF4-target genes in p30-expressing cells, making them more sensitive to acute ER stress, and conferring a new potential vulnerability to preferentially eliminate p30-expressing cells.

models, with a common impact on primary response genes linked to AP-1 transcription factors. However, the specific inflammatory genes affected vary by cell type, reflecting the cell type- and stage-specific nature of the inflammatory transcriptional program[68]. Longer LPS exposures preferentially damage p42-expressing cells, suggesting that the absence of a functional p42 in *CEBPA* mutant cells may exert a protective effect under chronic inflammatory conditions. This may result either from cell intrinsic pro-apoptotic effects triggered by long inflammatory exposures[44,79,80], or by the auto- and paracrine effects of the released cytokines. The bone marrow microenvironment undergoes severe alterations in AML, including inflammatory activation in stromal populations[81]. This leads to the functional decline of normal hematopoietic cells, but the mechanisms underlying the increased resistance of mutant cells remain unclear[82]. We demonstrate that the canonical CEBPA isoform, p42, predominantly expressed in myeloid progenitors, plays a significant role in the inflammatory response. We propose that the AML pro-inflammatory niche may generate a strong selective pressure that favors the expansion of cells that cannot express p42. Since CEBPA is an essential protein for myeloid progenitor cells, null mutations are likely to be counter-selected (with the exception of rare cases that accumulate homozygous C-terminal mutations). However, biallelic mutants–in which the only functional CEBPA is the p30 isoform – may be strongly favored, as p30 can maintain myeloid cell identity while reducing susceptibility to the pro-inflammatory microenvironment. Resistance to inflammation may influence leukemic evolution at different stages. In early stages, even pre-leukemic, either acute inflammatory episodes or chronic inflammation could act as a strong selective pressure, favoring the expansion of inflammation-resistant clones. Alternatively, resistance to inflammation may become more important at later stages, within a pro-inflammatory leukemic microenvironment, where defective inflammatory responses might allow CEBPA-mutant cells to evade the bone marrow inflammatory stress. Our findings align with a previous report showing that chronic interleukin-1 exposure selects for *CEBPA*-knock-out HSPCs[69], collectively supporting the idea that chronic inflammatory conditions may favor the outgrowth of *CEBPA*-mutant clones, as

observed with other common AML mutations[52–55,59,83,84]. Importantly, recent pre-clinical and clinical studies have aimed to target cell intrinsic and extrinsic inflammation as promising therapies in hematological malignancies with encouraging results[85,86].

Our findings indicate that a major role of CEBPA may involve regulating basal levels of inflammatory genes and establishing the appropriate transcription factor circuitry and enhancer predisposition to effectively respond to inflammatory signals. Typically classified as a lineage-determining TF[2], fine-tuning inflammatory responsiveness may be one of CEBPA's critical functions, akin to other lineage-determining myeloid TFs such as PU.1[87] or IRF8[88]. However, unlike these two factors, CEBPA's primary function in this context may lie in activating and cooperating with AP-1 factors, a key family of signal-responsive transcription factors (Fig. 8). The role of AP-1 members is intricate, as they serve as effector transcription factors for various signaling pathways, and their highly combinatorial nature allows them to impact diverse cellular functions[63,64,73,74]. A recent study revealed a dependency on FOS for the survival of *CEBPA*-mutant AML cells[89]. Interestingly, we observed a downregulation of FOS expression compared to patients carrying other AML mutations, indicating a need for tight regulation of FOS levels, as its expression is necessary for survival but excessive levels may be detrimental. Additionally, we found that unlike p42, p30 does not dimerize with ATF4, dampening the expression of their target genes (Fig. 8). This differential interaction underscores the importance of the N-terminal region in determining the specificity of the interactions with other bZIP proteins, warranting further mechanistic investigation. ATF4, known for its central role in the integrated stress response and requirement for HSC homeostasis, may thus represent a potential vulnerability to target *CEBPA*-mutant AML[71,72,90]. In line with this, we show that p30-expressing cells are more sensitive to tunicamycin-induced ER stress. Importantly, our conclusions only apply to AML, as CEBPA is not frequently mutated in other cancer types. Future investigations elucidating the interplay between CEBPA and different stress response pathways will be essential for identifying further vulnerabilities of *CEBPA*-mutant cells and exploring new therapeutic avenues.

## Methods

### Cell culture

HPC-7 cells were kindly provided by L. di Croce (CRG) and were cultured in Iscove's Modified Dulbecco's Medium (IMDM) (Gibco 12440-053) supplemented with 10% FBS, 1% penicillin/streptomycin (Gibco 15140-122), 1% L-glutamine (Gibco 25030-024), 50 μM b-mercaptoethanol (Gibco 31350-010), and 50 ng/mL murine SCF (Miltenyi Biotec, 130-101-697). KO-52 cells (JCRB0123) were cultured in Dulbecco's Modified Eagle Medium (DMEM) (Dominique Dutscher, L0106-500) supplemented with 10% FBS, 1% penicillin/streptomycin and 2% L-glutamine. For TLR-4 stimulation, cells were exposed to LPS (Escherichia coli O111:B4, Sigma L2630) at a final concentration of 100 ng/mL. To induce ER stress, cells were incubated with tunicamycin (Sigma, T7765) at 1 μg/mL for HPC-7 and at 2 μg/mL for KO-52 cells. To induce translocation to the nucleus in cells expressing ERT2-fusion proteins, cells were treated with 400 nM 4-hydroxytamoxifen (4-OHT) (Sigma, H7904) 24 h before the experiment, including the empty vector controls. Where indicated, cell viability was assessed by CellTiter-Glo luminescent assay (G7571, Promega). The assay was performed following manufacturer's instructions. 25,000 cells were initially plated and treated for the indicated time. Luminescence was measured after addition of Cell Titer-Glo reagent in a Synergy H1 luminometer, 1 s integration time. For methylcellulose plating, 1000 KO-52 cells were plated in 1.1 mL of MethoCult H4230 (Stem Cell Technologies) in 6-well plates. Three wells per condition were plated. Media was supplemented with 400 nM 4-OHT and LPS or tunicamycin where indicated. Colonies were manually counted on day 14. 1000 cells/well were used for replating. As described in ref. 69., to calculate the cumulative CFU potential and account for the overall clonogenic activity of KO-52 cells, we multiplied the total number of cells from the first plating by the CFUs formed per 1000 cells plated at replating[69].

### Mouse bone marrow culture

All mouse experiments were conducted according to protocols approved by the Danish Animal Ethical Committee, the Danish Animal Eperiments Inspectorate and the Department of Experimental Medicine at the University of Copenhagen with regard to the three R's (refine, reduce, and replace) of animal experiments. Animals were housed in individually ventilated cages, in a temperature- and humidity-controlled room with a 06:00–18:00 h light cycle and fed a standard chow diet and tap water *ad libitum*. All experiments were carried out under supervision of veterinarians of the Department of Experimental Medicine, University of Copenhagen. The genotypes used were *Cebpa*$^{Fl/p30}$; *R26-Cre-ER* and *Cebpa*$^{Fl/Fl}$. Mice were backcrossed for at least eight generations onto the C57BL/6 background[38,91]. Hematopoietic cells were harvested from the bones (tibia, fibia, femur, ilium) of 13-16 weeks old mice and isolated by crushing in PBS + 3%FBS and were subsequently frozen in FBS + 10% DMSO.

Total bone marrow cells were stained with a cocktail of biotin-conjugated lineage antibodies: Nk-1.1 (clone PK136, 108703), CD3 (clone 145-2C11, 100303), CD4 (clone RM4-5, 100507), CD8 (clone 52-6.7, 100703), CD11b (clone M1/70, 101203), CD19 (clone 6D5, 115503), Ter119 (clone TER-119, 116203), Ly6G/Ly6C (clone RB6-8C5, 108403) and B220 (cloneRA3-6B2, 103203), all from BioLegend. Lineage positive cells were depleted using EasySep Mouse Streptavidin RapidSpheres (Stem Cell 5001), and lineage-negative cells were cultured in IMDM 20% FBS, 1% P/S, 1% L-Gln, 50 μM b-mercaptoethanol supplemented with 100 ng/mL SCF, 100 ng/mL IL-6 (Peprotech 216-16), 100 ng/mL Flt3 (Miltenyi biotec 130-094-038) and 10 ng/mL IL-11 (Stem Cell 78026.1). Cells were plated in three biological replicates at 100,000 cells/100 μL with 400 nM 4-OHT for three days, and then stimulated with 100 ng/mL LPS. For methylcellulose plating, 10,000 lineage-negative cells were plated in 1.1 mL of MethoCult M3534 (Stem Cell Technologies) in 6-well plates. Media was supplemented with

400 nM 4-OHT (for both genotypes) and LPS was added at 100 ng/mL. Colonies were manually counted on day 10. Three wells per biological replicate and condition were plated.

To generate bone marrow-derived macrophages, bone marrow cells were cultured in complete DMEM medium (10% FBS, 1% penicillin/streptomycin, 1% L-glutamine, 1 mM sodium pyruvate) with 20% L929-conditioned media. To induce Cre nuclear translocation in lineage-negative cells or in macrophages, 400 nM 4-OHT (Sigma, H7904) was added in the media, both in control (wild-type) and p30 cells. Macrophages were stimulated with 100 ng/mL LPS on day 9.

### Transductions and transfections

To generate p42 HPC-7 and p30 HPC-7, specific primers were used to amplify the two different CEBPA isoforms from donor plasmids. Amplicons were then inserted into a pMSCV-IRES-GFP empty backbone by Gibson assembly together with the ER$^{T2}$ fragment. Platinum-E cells were used to generate retroviral particles containing the CEBPA-p42-ERT2 and CEBPA-p30-ERT2 plasmids. Transfection was performed by adding 20 μg of DNA, 0.3 mM CaCl$_2$ and then 500 μL of HBS 2X. Supernatant containing virus was collected 48 h after transfection, filtered through a 0.45 μm filter and used directly on HPC-7 cells supplemented with 8 μg/mL polybrene and centrifuged at $1000 \times g$ for 1h30min. Cells were later sorted for GFP in a FACS Aria II (BD Biosciences). For constitutive expression of CEBPA in HPC-7, cells were transduced with pMSCV-p30-flag-IRES-GFP and pMSCV-p42-flag-IRES-GFP plasmids from VectorBuilder. To knock-down *Atf4* in HPC-7 cells, plasmids containing shRNAs for scramble sequence and targeting *Atf4* were kindly provided by S. Sykes Lab (WUSTL) and cells were sorted for RFP657. FOS overexpression in KO-52 cells was achieved by transfecting pLX-FOS plasmid (Addgene #59140) alongside psPax2 and VSVG into HEK293T cells by the calcium phosphate method. Supernatant containing the lentiviral particles was collected 48 h after transfection, filtered, supplemented with 8 μg/mL polybrene and used on KO-52 cells by centrifugation, cells were later selected with 10 μg/mL blasticidin (BLL-44-02, InvivoGen). For long-term CEBPA expression, KO-52 cells were transduced with p42-ERT2 and p30-ERT2 plasmids and sorted for GFP. For CEBPA transient expression in KO-52, $3 \times 10^6$ KO-52 cells were electroporated with 2 μg of plasmid: p42-FLAG-GFP, p30-FLAG-GFP or the GFP empty plasmid with a Nucleofector 2b (Lonza) using program V-001. GFP+ cells were sorted 24 h after transfection. 24 h post-sorting, cells were treated with 100 ng/mL LPS. For co-immunoprecipitation experiments, plasmids constitutively expressing human p42-FLAG-GFP, p30-FLAG-GFP, WT-CEBPA-FLAG, CEBPA-AroPerfect-FLAG, CEBPA-AroLite-FLAG and CEBPA-ΔTEIII-FLAG were synthesised at VectorBuilder and were transfected into HEK293F cells and then treated with 2 μg/mL tunicamycin for 4 h to induce ATF4 translation. The sequences of AroLITE and AroPERFECT mutants were obtained from previous publications[75,76].

### Flow cytometry

To quantify cell death, cells were stained with Annexin V-APC (BD Pharmingen 550474) and DAPI. Differentiation stage was assessed with the following antibodies: CD11b-PE/Cy7 (clone M1/70, BioLegend 101215), F4/80 - PE (clone T45-2342, BD Pharmingen 565410), Gr1 (Ly6G/C)-biotin (clone RB6-8C5, BioLegend 108403), human CD14-APC (130-091-243, Miltenyi biotec). PE-Streptavidin (BioLegend 405203) was used to detect the biotin-bound antibody. All antibodies were used at a 1:400 dilution in PBS-FBS 2%. Cells were analyzed on a FACS Canto II (BD Biosciences) and data analyzed on FlowJo v10.6.2 software.

### Supernatant collection for secretome identification and cytokine array

p42 and p30 HPC-7 cells were washed and plated in IMDM without phenol red (Gibco, 21056-023) with 10 ng/mL SCF, 400 nM 4-OHT and

exposed to 100 ng/mL LPS for 0 h, 8 h or 16 h. Supernatants were collected by 2-step centrifugation, first at $350 \times g$ to remove cells followed by 5 min centrifugation at $2000 \times g$ to remove debris. For cytokine arrays, cells were plated in standard supplemented media (see section 'Cell culture') and the supernatants were collected and filtered through a 45 µm filter. The *Proteome Profiler Array, Mouse Cytokine Kit Panel A* (ARY006 R&D Systems) was used following manufacturer's instructions and imaged with an Odyssey CxL instrument (LI-COR).

### Secretome identification by mass spectrometry
Supernatant sample purification and preparation was performed as previously described[92]. 50% of supernatant volume TRIS-HCl (pH 8) with TCEP 10 mM CAA 55 mM was added, and samples heated to 95 °C for 10 min. Acetone was added to media (80% v/v) and incubated overnight at −20 °C to precipitate proteins. Proteins were digested in 8 M Urea with LysC, and with 50 mM ABC and Trypsin. Peptides were desalted on reversed-phase C18 stage times and eluted with 80% (v/v) acetonitrile in 0.5%(v/v) acetic acid. Eluants were dried in a SpeedVac and resuspended in 2% acetonitrile (v/v) and 0.1% TFA (v/v). Samples were measured by liquid chromatography-tandem mass spectrometry using Vanquish chromatographic system (Thermo Fisher Scientific) and the Exploris 480 mass spectrometer (Thermo Fisher Scientific). Peptides were separated by 90-min chromatographic gradients using a binary buffer system with buffer A (0.1 % formic acid in LC-MS-grade water) and buffer B (80% ACN, 0.1% formic acid in LC-MS-grade water), with an in-house packed 50 cm analytical column. Samples were measured in data-independent acquisition (DIA) mode with a window m/z range from 400 – 1000 m/z separated into 25 isolation windows with a size of 24 m/z per window. We used a staggered window approach with isolation windows shifted by 12 m/z. We measured pooled samples to create a gas phase fractioned (GPF) spectral library where similar LC-MS methods were applied, only the DIA scan ranges were limited to 100 m/z covering the overall m/z range from 400 to 1000 m/z in six consecutive runs and staggered DIA windows with a size of 4 m/z.

### Western blot
Cells were collected, washed in PBS, lysed in Laemmli buffer and boiled before loading the samples into 10% resolving and 4% stacking polyacrylamide gels. Running was performed at an intensity of 25 mA/gel and 180 V. Gels were transferred to nitrocellulose membranes (Amersham Protran, GE 10600002), at 350 mA, 100 V for 1 h on ice. Membranes were incubated overnight at 4 °C with the following primary antibodies: CEBPA (D56F10 8178, Cell Signaling), FOS (T.142.5 MA5-15055, Thermo Fisher), ATF4 (B3 118155, Cell Signaling), FLAG M2 (F1804, Sigma), ERα (F-10 sc-8002, Santa Cruz Biotechnology), H4 (ab10158, abcam), α-tubulin (B-5-1-2 T6074, Sigma) and GAPDH (6C5 sc-32233, Santa Cruz Biotechnology). Odissey CxL scanner (LI-COR) was used to visualize the protein. The images were analyzed and quantified using ImageStudioLite v5.2.5 (LI-COR software). For quantification, intensity of the band of interest was normalized to α-tubulin or GAPDH signal. For fractionation, cells were initially lysed in 10 mM Tris pH7.8, 10 mM KCl, 1.5 mM MgCl$_2$, 1 mM DTT and protease inhibitors and incubated 10 min on ice, after centrifugation at maximum speed, 4 °C for 1 min, supernatant corresponding to the cytoplasmic fraction was collected. After a wash, pellets were then further lysed in 10Mm Tris pH7.8, 0.42 M NaCl, 1.5 mM MgCl$_2$, 0.2 mM EDTA, 25% glycerol, 1 mM DTT and protease inhibitors and incubated on ice for 30 min. Samples were centrifuged at maximum speed for 1 min at 4 °C and supernatant was collected as nucleoplasm and the pellet, containing the chromatin fraction, was resuspended in Laemmli buffer and sonicated.

### RT-qPCR and RNA-seq
RNA was isolated from $1-2 \times 10^6$ cells in 1 mL of TRIzol Reagent (ABP Biosciences FP312) following the manufacturer's instructions. For low

starting cell number samples, RNA was extracted with Direct-zol RNA MicroPrep kit (Zymo R2060). The RNA-seq of KO-52 overexpressing CEBPA was normalized using ERCC RNA Spike-In controls (4456740, LifeTechnologies). For retrotranscription, cDNA synthesis was performed from 1 µg of RNA using the High-Capacity RNA-to-cDNA Kit (Applied Biosystems 4387406) and qPCRs using the PowerSYBR Green PCR Master Mix (Applied Biosystems 4367659) in a QuantStudio 7 Flex instrument (Applied Biosystems). Ct values were normalized to *RPL38* and *HPRT or Rpl32*. A list with all the primers used is shown in Supplementary Table 2. Strand-specific mRNA-seq libraries were prepared and sequenced.

### ATAC-seq
ATAC-seq was performed from 50,000 nuclei per replicate using an in-house synthetized Tn5 and loaded with standard Illumina compatible adaptors, 30 min, 37 °C. DNA was purified by Zymo ChIP DNA Clean and Concentrator Kit. Transposed fragments were amplified with NEBNext High-Fidelity PCR Master Mix (NEB M0541). Libraries were cleaned and size-selected using CleanNGS beads (CNGS-0050) and quality was assessed by Tapestation and Qubit.

### ChIP-seq
HPC-7 cells were cross-linked with 1% formaldehyde and lysed (1% Triton, 0.1% sodium deoxycholate, 0.1% SDS, 0.2 M NaCl, 10 mM Tris pH 7.5, 10 mM EDTA and 1X protease inhibitor cocktail (Roche 11836170001)). Sonication was performed in the same buffer with 0.5% SDS in a Bioruptor (Diagenode) for 60 cycles (30 s ON 30 s OFF), vortexing every 20 min. Sodium butyrate 5 mM was added to the buffers in the cells for the H3K27ac ChIP to inhibit deacetylases. For H3K4me1 ChIP-seq, DNA was digested in 50 mM Tris pH 7.5, 4 mM MgCl$_2$ and 2 mM CaCl$_2$ with 10U micrococcal nuclease (Thermo Scientific EN0181) for 15 min at 22 °C. For ATF4 ChIP, KO-52 cells were pretreated with 2 µg/mL tunicamycin for 4 h and additionally crosslinked with 1 mM DSG (A35392, ThermoScientific) for 45 min. Lysates were sonicated for 30 min in lysis buffer (0.1% SDS). Lysates were incubated with 4 µg of H3K27ac antibody (Abcam ab4729), 8 µg of anti-ERα (F-10 sc-8002X, Santa Cruz Biotechnology), 3 µg anti-H3K4me1 (Diagenode, C15410194), 0.5 µg anti-CEBPA (D56F10 8178, Cell Signaling) or 0.7 µg anti-ATF4 (B3 11815, Cell Signaling) overnight at 4 °C. Antibody-bound chromatin was pulled down with protein G Dynabeads (Invitrogen 10003D). Samples were washed, RNase-treated and reverse cross-linked by incubation at 65 °C in 1% SDS, 0.1 M NaHCO$_3$ and proteinase K. DNA was purified using ChIP DNA Clean & Concentrator Kit (Zymo D5205). Libraries were prepared using the NEBNext Ultra DNA Library Prep kit (E7645, New England Biolabs).

### HiChIP
HiChIP was performed in duplicates using Dovetail Genomics' HiChIP *MNase* kit and following manufacturer's instructions. Briefly, $5 \times 10^6$ cells were pelleted to then be crosslinked, digested with *MNase* and lysed. After quality control, 1000 ng of lysate were incubated with H3K27ac antibody (D5E4 Cell Signaling, 8173). Samples were then proximity ligated and DNA was purified for library preparation, followed by ligation capture and amplification. Final DNA was quantified and profiled in a Bioanalyzer.

### Co-IP
Cells were collected by centrifugation. Cellular pellets were washed twice in PBS and re-suspended in RIPA buffer supplemented with protease inhibitors and benzonase (Sigma-Aldrich). Lysates were incubated for at least 6 h at 4 °C on a rotating wheel and centrifuged at $17,000 \times g$ for 15 min to remove cellular debris. Input was taken and 20 µL of anti-FLAG® M2 Affinity gel (A2220, Sigma-Aldrich) was added for each condition for overnight incubation. The beads were then washed 4 times in RIPA buffer to remove non-specific protein bound

and re-suspended in 40 μL of 1X Laemmli buffer containing glycerol with 50 mM dithiothreitol (DTT) and bromophenol blue. The samples were then boiled at 95 °C for 5 min, centrifuged at maximum speed to pellet the beads and the supernatant was used for western blot analysis against ATF4 (Cell Signaling 11815) and Flag M2 (Sigma F1804).

## Statistics and reproducibility

All experiments were done in triplicate, three biological replicates (i.e. number of mice, number of individual experiments for a cell line) per condition, including all RNA-seq experiments. H3K27ac and H3K4me1 ChIPs were also performed in triplicates; ATF4 and CEBPA ChIP-seq using ER and CEBPA antibodies, one replicate per condition. ATAC-seq and H3K27ac HiChIP were done in duplicates. Mass spectrometry experiments were performed with four replicates. Co-immunoprecipitation experiments were independently reproduced twice. Measurements were taken from distinct samples, not the same sample measured repeatedly. Two-group comparisons were analyzed using unpaired two-tailed t test, p values < 0.05 were considered statistically significant. For mice experiments comparing the same biological sample, t tests were paired. Data was analyzed using Graph Pad Prism v.10.4.1 and shown as mean +/− SEM unless otherwise stated. Center line in boxplots represents the median and violin plots show median and quartiles. Adjustment for multiple comparisons was done using Benjamini-Hochberg tests. Correlation coefficients were done using Pearson correlation tests. To calculate the significance of the overlap between different sets of genes we performed a permutation test with 100,000 random independent samplings of the same size as the observed gene sets to model the mean and standard deviation of random overlapping which followed a normal distribution. Then the p value was calculated for the observed overlaps.

## RNA-Seq analysis

RNA-seq raw sequencing reads were aligned to the human or mouse reference genome (hg38 and mm39 assemblies respectively) with bowtie2 v2.4.4 with the "--very-sensitive" tag[93]. Quantification of gene expression was performed with *featureCounts* from Subread v2.0.3 using uniquely mapping reads on NCBI *RefSeq* gene annotation for either human hg38 or mouse mm39[94]. Differential gene expression analysis was done in R using the DESeq2 package[95]. Spike-in normalization to account for differences in library size was done using the estimateSizeFactors function in DESeq2. For public *CEBPA*-mutant AML RNA-seq data (TCGA and BEAT databases), differential gene expression analysis of already aligned raw counts was done using the R package *limma* and the "voom" method[96]. Databases were merged and batch normalized using the R package sva to remove technical differences between the two databases[97]. In all cases, genes were considered to be differentially expressed if their FDR was smaller than 0.05. Normalized counts were obtained also with DESeq2 using the *rlog* normalization method. GSEA was done using the clusterProfiler package in R including terms from the gene ontology biological process collection[98]. GO term enrichment of specific gene sets was done using *Enrichr*[99].

## ChIP-Seq and ATAC-seq analysis

ChIP-seq reads were pre-processed with TrimGalore v0.6.6 for quality and adapter trimming. Trimmed reads were aligned to mouse genome mm39 or human hg38 genome using Bowtie2[93] v. 2.4.4 with '--very-sensitive' flag set and a minimum fragment length (-X flag) of 1000 bp, all other parameters set to default. Duplicate reads were identified using Picard MarkDuplicates and excluded from the downstream analysis. Genome-wide normalized coverage tracks in bigwig format were generated using DeepTools[100] bamCoverage v. 3.3.1. ChIP-seq peaks were identified by MACS2[101] v. 2.2.5 for individual replicates and pooled replicates for each condition, using input libraries as control. Peaks identified in each condition were merged with bedtools[102] merge

(v. 2.31.1) to generate a common peak set. Reads from each sample were quantified over the common peak set using the regionCounts function from the csaw[103] R package and differential binding analysis was performed with DESeq2[95]. The murine ATF4 ChIP-seq peaks were computed from the dataset GSM881130; CEBPA and FOS peaks in KO-52 were computed from GSE211095; and CEBPA ChIP-seq in GMPs from GSE118963.

## HiChIP analysis

HiChIP data were analyzed following the Dovetail Genomics HiChIP guidelines. In brief, reads were mapped to the mm39 genome with the BWA-MEM software (v. 0.7.17) with specific settings to allow mapping of ligation events (bwa mem -5SP). Alignment files were further processed with the 'pairtools' software to identify and record valid ligation events (pairtools parse) and to remove PCR duplicates (pairtools dedup). Quality of HiChIP libraries was confirmed by observing the expected fractions and insert size distributions of valid trans and cis non-duplicated pairs (libraries were considered of good quality when the count of non-duplicated cis pairs of more than 1000 bp was greater than 20% of the total mapped non-duplicated pairs). Also, enrichment of HiChIP reads at target protein's binding sites previously defined by peak calling of ChIP-seq data was observed, as expected. Loop calling (i.e. identification of significant interactions between each target protein and the surrounding genome) and differential loop analysis were performed with FitHiChIP[104]. Contact maps were generated with juicer tools[105]. Loops and read coverage at regions of interests were plotted with custom scripts using the sushi R package[106].

## Virtual 4 C

HiC data at 5 kb resolution from AML patients CEBPA mutant and CEBPA WT from ref. [66]. and HiChIP data in HPC-7 were converted from.cool to.bedpe using the R package HiCcompare[107]. Then the genomic interactions contacting the indicated promoter region were extracted with the R package GenomicInteractions[108]. The mean normalized signal (reads per million) per each group was calculated for plotting.

## LC-MS/MS analysis

Staggered windows were deconvoluted with the MSConvert tool of the ProteoWizard software suit[109] (v. 3.0.21321). Spectral library generation and peptide identification/quantification from LC-MS raw data was performed with the DIA-NN software suit[110] (v. 1.8). We used the SWISS-PROT *Mus musculus* fasta database downloaded from UniProt (v. 2021-11-18) to make a spectral library using the six GPF measurements from pooled samples. Trypsin was set as the digestion enzyme with a maximum of one miss-cleavage and cysteine carbamidomethylation was set as a fixed modification. The scan window radius was set to 10, mass accuracies were fixed to 2e-05 (MS2) and 7.5e-06 (MS1), respectively. Precursor peptides were filtered at an FDR < 1%. Label-free normalization of protein groups was performed in R using the MaxLFQ algorithm[111] and proteotypic peptides only.

## Reporting summary

Further information on research design is available in the Nature Portfolio Reporting Summary linked to this article.

# Data availability

The data generated in this study has been deposited in NCBI's Gene Expression Omnibus under GEO Series accessions GSE270547, GSE270549, GSE288477 and GSE270550. The mass spectrometry proteomics data have been deposited to the ProteomeXchange Consortium via the PRIDE partner repository with the dataset identifier PXD053800. The following previously published datasets were used: ATF4 ChIP-seq peaks were computed from the dataset GSM881130, CEBPA and FOS peaks in KO-52 were computed from GSE211095,

CEBPA ChIP-seq in GMPs from GSE118963 and the human AML HiC data from GSE152136. Source data are provided with this paper.

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

## Acknowledgements

We thank all the Cuartero lab members for helpful discussions, Clara Berenguer for technical advice, and the IGTP Flow Cytometry and Genomics core facilities. We sincerely thank Stephen Sykes and Mary Basse (Washington University in St. Louis) for generously providing the plasmids used for Atf4 knockdown. The Cuartero lab acknowledges funding by 'La Caixa' Foundation and from the European Union's Horizon 2020 research and innovation programme under the Marie Skłodowska-Curie grant agreement No 847648 (fellowship code LCF/BQ/PI20/11760002); the Spanish Ministry of Science and Innovation (PID2020-117950RA-I00 and PRE2021-097862); AGAUR; the American Society of Hematology (ASH) Global Research Award; and the Josep Carreras Leukaemia Foundation. G.M.-C. and L.L were supported by the European Union's Horizon Europe Research and Innovation program 2021-2027 under the Marie Skłodowska-Curie Actions Grant Agreements n°101081347 and 101068212, respectively. Work in the Porse lab was supported through a center grant from the Novo Nordisk Foundation (Novo Nordisk Foundation Center for Stem Cell Biology, DanStem; Grant Number NNF17CC0027852) and performed in the context of the Danish Research Center for Precision Medicine in Blood Cancers funded by the Danish Cancer Society (R223-A13071) and Greater Copenhagen Health Science Partners. Work in the Merkenschlager lab was supported by the Medical Research Council. Work in the Vaquero lab was supported by the Spanish Ministry of Science and Innovation (PID2020-117284RB-I00).

## Author contributions

Conceptualization and study design: M.C.-F., S.C. Experiments: M.C.-F., S.C., J.-M.C-G., F.D.W., A.-K.F., E.J.-V., C.P.M.C., L.Y., A.G.-G. Data analysis: M.C.-F., G.M.-C., L.L., F.D.W., Y.-F.W. Resources: A.-K.F., B.P., F.D.W. and A.-K.F. contributed equally. Supervision: A.V., F.M., M.M., B.P., S.C. Writing and figures: M.C.-F., G.M.-C., S.C.

## Competing interests

The authors declare no competing interests.
