## [Transparent Peer Review file · Nature Communications]

Mutant CEBPA promotes tolerance to inflammatory stress through deficient AP-1 activation

Corresponding Author: Dr Sergi Cuartero

Version 0:

Reviewer comments:

Reviewer #1

(Remarks to the Author)

Cadefau-Fabregat et al. investigates the impact of the CEBPA mutant form p30 in acute myeloid leukemia. They demonstrate the first time that expression of the p30 mutant AML cells have impaired inflammatory gene expression and altered response to LPS. They provide substantial mechanistic insights, by demonstrating the importance of the AP-1 transcription factor family as critical downstream targets of CEBPA in AML. The study is highly comprehensive, with many different models and methods, and offers important novel insights into the role of the CEBPA mutant in AML. The work is overall well done and solid. However, I am not fully convinced about the interplay between CEBPA and ATF4.

Major points

- 1) The authors used first a rather artificial system via tamoxifen induction of ER-fusion proteins. This approach can sometimes suffer from leakiness. As control, the authors should demonstrate that both proteins translocate from the cytoplasm into the nucleus and bind to chromatin upon 4-OHT induction.
- 2) There is no explanation provided why p42, but not p30, is interacting with ATF4. Both proteins possess the dimerizing bZIP domain and it appears illogical why p30 cannot interact with ATF4. AlphaFold 3 predicts an interaction for both p30 and p42 with ATF4. Thus, some more mechanistic insights would strengthen the manuscript.
- 3) The authors propose that p42 particularly cooperates with ATF4 for gene regulation. To make this statement stronger, I would recommend deleting ATF4 from the HPC-7 cells. One would expect weaker differences upon tamoxifen induction of p42 versus p30. This could be assessed by RT-qPCR of key dysregulated genes from Figure 1A.

Minor Points:

- 1) line 378: p42-expressing > p30-expressing
- 2) lines 455-458: Where is this result shown?
- 3) Figure 7B should also include the ATF4 ChIP-Seq signal.
- 4) it would be nice to present a figure, that summarizes the main points of the manuscript
- 5) The title should make clearer that the results refer to AML. No data are shown that the conclusion is more general, beyond AML.
- 6) The Figure legends for Figure 4D should make clearer that the shown data are mass-spectrometry data.
- 7) The authors used publicly available ATF4 ChIP-seq data, but it is not fully clear how the data were used. I would recommend to explain this point better. Also why did the authors use two different sets of ATF4 ChIP-Seq data, and not just the better one?

Reviewer #2

(Remarks to the Author)

In this manuscript, Cadefau-Fabregat et al, investigate the differential transcriptional and gene regulatory consequences of canonical p42 and AML-associated p30 isoforms of CEBPA. Whilst DNA binding and transcriptional activation are broadly unchanged, the authors identify a selective defect in activation of AP1-dependent inflammatory and early-response genes. Through analysis of HPC-7 cells with inducible expression of p30, complemented by conditional *Cebpa*^{Fl/p30}; R26-Cre-ER

BM progenitors and primary macrophages, and p42 rescue of KO-52 cells, the authors show that unique expression of the p30 isoform affects acute inflammatory gene expression including molecular and cellular responsiveness to LPS by interfering with the strength of enhancer activation at regulatory regions also bound by AP-1 family members. The weaker and delayed inflammatory effects are partially mediated by Fos downregulation and may reduce pro-apoptotic effects of the inflammatory process, although this is not evident in primary cells. The authors suggest that the relative gain in fitness of p30-expressing cells upon prolonged inflammation may facilitate the malignant process, but there is limited or no data to support this assertion beyond the unchanged apoptosis and higher expansion of HPC-7 p30+ cultures in the presence of LPS and unchanged colony formation by p30 BM progenitors. In a related, but separate mechanistic exploration, the authors also show a specific effect of p30 in ATF4-mediated stress responses, which they suggest may be explored as a therapeutic mechanism. Specifically, p30, unlike p42, does not interact with ATF4 (by co-IP), resulting in reduced transcriptional activation at CEBPA/ATF4 sites. In this case, ER stress leads to increased apoptosis in less responsive p30-expressing HPC-7 cells, highlighting a potential drug vulnerability of CEBPA mutant AML cells.

The study is well designed and well executed and the conclusions as to the differential transcriptional regulatory consequences of p42 and p30 isoforms of CEBPA are justified within the context of the cell models used. However, it is not clear to me which of these mechanisms apply to the leukemia process, as all the associations with AML patient samples are correlative on gene expression profiles that may to a large extent reflect cell composition, with no functional validation. Given the centrality of p30 expression to CEBPA mutant leukemia, this should be addressed directly to support the wider implications of the data. HPC-7 cells or differentiated macrophages are not representative of CEBPA mutant leukemia cells.

MAIN POINTS

1. The Authors should investigate the contribution of diminished AP-1 driven inflammatory gene transcription to CEBPA-mutant leukemia by:
 - a. determining the impact of LPS treatment on p30-transformed cells, either in vitro through serial replating, or, if feasible, in vivo through leukemia latency in the presence of a chronic inflammatory stimulus; conversely, the role of FOS down-regulation in dampening inflammatory response and facilitating transformation can be assessed through up-regulation of Fos.
 - b. inspecting p30, AP-1 gene family and H3K27ac chromatin binding at inflammatory and control loci in vitro (or in vivo)-transformed cells
 - c. measuring p30 and ATF4 chromatin binding at stress loci and measuring responsiveness of p30-transformed cells, in vitro or in vivo, to ER stress.
2. To consolidate the co-regulation of p30-target loci on AP1 factors, the Authors should perform ChiP-seq analysis of AP1 family, including ATF4. binding at p30 target loci with differential transcription.

MINOR POINTS

1. The Authors should clarify the significance of inducing p30 expression in differentiated macrophages to study the impact of the respective isoform on inflammatory gene expression. What is their abundance and relative inflammatory gene expression in CEBPA mutant leukemia?
2. How do the Authors position the defective AP-1-mediated inflammatory and ER stress responses in terms of their contribution to CEBPA AML? Is there cell type specificity, and how does a possible cell specificity CEBPA mutant leukemia mice compare to WT? Or do the Authors think that they may have a differential contribution at different stages in leukemia initiation and progression?
3. Can the Authors please clarify the nature of the progenitor colonies quantified in 4E upon LPS treatment of WT and p30 progenitors. This is important to understand the cellular effects of chronic inflammation in the context of p30 unique expression.

Reviewer #3

(Remarks to the Author)

The transcription factor CEBPA is frequently mutated in acute myeloid leukemia (AML), resulting in the production of a shorter isoform known as p30. Both the canonical 42-kDa isoform (p42) and the AML-associated p30 isoform bind chromatin and regulate transcription. However, the specific transcriptional programs governed by each isoform and their connection to the selective advantage in AML remain poorly understood. The authors focus on this knowledge gap, particularly the unclear mechanisms by which the differential gene regulation by p30 confers a selective advantage in CEBPA-mutant AML. To address this, they aimed to identify novel transcriptional programs differentially controlled by the CEBPA isoforms and to elucidate the molecular mechanisms driving these distinct transcriptional outcomes. Their findings indicate that p30 fails to promote the normal expression of inflammatory pathways, resulting in a diminished ability to mount an effective inflammatory response to LPS. This deficiency, they conclude, grants p30-expressing cells increased resistance to the harmful effects of prolonged inflammatory signals. Mechanistically, these differences are attributed to the differential regulation of AP-1 family proteins. The authors suggest that these discoveries establish a novel link between mutant CEBPA, inflammation, and the stress response.

To investigate the distinct functions of p42 and p30, the authors utilized tamoxifen-based experimental models. Initially, they used CEBPA transgenes fused to an estrogen receptor binding domain (ERT2) fragment, allowing them to control the nuclear entry of p42 and p30 with 4-hydroxytamoxifen (4-OHT). Through these constructs, they observed differences in inflammatory signaling between p30 and p42. In subsequent experiments, they employed mouse models in which p42 was deleted using Cre-ERT2.

A critical point to consider is that tamoxifen itself modulates various pathways, including the activation of endogenous estrogen receptors (ER) and AP-1 (PMID: 21233487, 29467493). Therefore, it is plausible that the observed transcriptional differences could stem from tamoxifen-activated ER and AP-1 interacting differently with the p42 and p30 isoforms in terms of binding affinities and protein interactions. The experimental design appears to lack the necessary controls to address these possible interactions properly. Notably, it seems that the empty vector was not treated with 4-OHT, and furthermore instead of an empty vector, the proper control should have included the ERT2 fragment without p42 or p30, treated with 4-OHT. Additionally, wild-type (WT) mice do not appear to have been treated with 4-OHT, further indicating that tamoxifen-related effects have not been adequately controlled.

Another concern arises from the initial experiments where the CEBPA isoforms were expressed as fusion proteins with the ERT2 fragment in cell models. The fusion of ERT2 to p42 and p30 could alter their DNA-binding properties and protein interactions, potentially to different extents between the isoforms, that could contribute to the transcriptional changes observed in inflammation-related pathways.

The authors noted that the transcriptional changes associated with p30 and p42 had not previously been linked to inflammation, despite extensive studies on these proteins. This is surprising because significant differences in inflammatory pathways would likely have been easily identified if they were a major factor between the isoforms. This prompts caution in interpreting the results from the specific models used in the study. In summary, the primary concern is that the inferred differences in inflammatory responses between p30 and p42 may be primarily driven by tamoxifen-induced effects. This concern is a recurring theme in the points raised below.

Questions/ Points:

1. What are the gene expression changes in HPC-7 cells if the experiments were conducted using constitutively expressed plasmids for p42 and p30 instead of tamoxifen-inducible ones? Specifically, how significantly are inflammatory pathways changed under these conditions?
2. Supplementary Fig. 1B is missing data for differentiation in control cells. Specifically, what are the expression levels of GR1 and CD11b in HPC7 cells with the control vector, both with and without tamoxifen induction?
3. How many of the genes overexpressed by p42 in Fig. 1A are downregulated in Fig. 1C following p42 deletion via Cre-ERT2?
4. Given the concerns about tamoxifen, how would the inflammatory gene expression levels compare if the authors used an MX-Cre mouse model instead of Cre-ERT2 to delete p42 in macrophages and progenitors, as shown in Figures 1C-F?
5. Do inflammatory genes show significant differences when comparing data from previously published models, such as Cebpa^{p30/p30} HSCPs or GMPs versus wild-type HSCPs or GMPs?
6. The gene expression findings in KO-52 cells following p42 overexpression stand in contrast with the effects observed in HPC-7 cells and the mouse model, as shown in Fig. 1A, C, and E. In the latter models, p42 overexpression resulted in a balanced number of upregulated and downregulated genes. However, in Fig. 2E, p42 overexpression predominantly upregulates genes. Could the authors explain this discrepancy and why p42 overexpression in Fig. 2E leads to mainly upregulated genes, while in Fig. 1, it shows a more balanced impact on gene expression?
7. Also related to the previous point, in KO-52 cells overexpressing p42, more than 2,600 genes were upregulated, as shown in Fig. 2E, while only a small fraction of genes were downregulated. This suggests a significant skew in the data, with some genes showing more than a 10-fold upregulation on a log₂ scale (which translates to over 1,000-fold changes on gene expression on a linear scale, a rather unphysiological result). To address this apparent bias in expression patterns, the authors should consider running external RNA spike-in controls if they have not already done so.
8. Given the substantial transcriptional differences induced by p42 overexpression in KO-52 cells (as shown in Fig. 2E), the changes in inflammatory gene expression observed with p42 overexpression in Fig. 2G seem relatively minor. If normalized to the empty vector (EV) control, the fold changes caused by p42 would likely be minimal compared to the over 2600 genes that are strongly upregulated by p42, as shown in Fig. 2E. These findings suggest that inflammatory genes are only weakly affected by p42 overexpression in the absence of a tamoxifen-inducible model.
9. While KO-52 is a CEBPA mutant case, plasmid-based overexpression of transgenes can lead to unnaturally high levels that may activate various stress pathways. Therefore, proper controls should include the overexpression of p30 in KO-52 cells. Specifically, what are the expression changes when p30 is overexpressed in KO-52 cells compared to empty vector (EV) controls, particularly regarding AP-1 family members and inflammatory genes?
The authors should include analogous experiments to those shown in Fig. 2E-2I, but with p30 overexpression in KO-52 cells. Do the inflammatory genes highlighted in Fig. 2H remain low or become further downregulated when p30 is overexpressed in KO-52 cells, as assessed using GSEA approaches?

10. The inflammatory AML module, downregulated in CEBPAbi/NT-mutated AML, was upregulated following p42 transfection (Fig. 2H). How are these inflammatory genes regulated when p30 is overexpressed?
11. Does p42 overexpression in KO-52 restore LPS response compared to p30 overexpressing KO-52 cells?
12. What are the gene ontology results for the 17% of overexpressed genes in p30 HPC-7 cells (Fig. 3D) and p30 macrophages (Fig. 3G) after LPS induction? Additionally, how many of these genes overlap between HPC-7 cells and macrophages?
13. What are the GSEA scores for the Hallmark Inflammatory Response (used in Fig. 2A) and the Inflammation Module (used in Fig. 2H) when comparing p30- versus p42-expressing HPC-7 cells after LPS stimulation at 2 and 12 hours?
14. The authors state that remarkably, a significant overlap in LPS-responsive genes was observed (Fig. 3H). However, only 30 overlapping genes were identified, while nearly 2,000 genes are differentially regulated between macrophages and HPCs, and about 500 more in KO-52 cells. This limited overlap does not strongly suggest a conserved inflammatory response. Additionally, it is unclear how many of these 30 genes might be detected by random chance. The authors should include a false discovery rate corrected p-value to determine if these 30 genes are statistically significant.
15. While CXCL10 and potentially GPR84 among the conserved 30 genes are associated with inflammation, a brief review of Sod2 and Serpine1 suggests only a weak connection to inflammation, if any. Can the authors explain how relevant these 30 genes are to the immune phenotype?
16. It is puzzling that the authors do not present the expression levels of AP-1-related genes in the LPS-treated experiments shown in Fig. 3, given that this is central to the study. What are the expression changes of AP-1-related genes in p30 and p42 HPC-7 cells following LPS stimulation at 2 and 12 hours?
17. The LPS-stimulated inflammatory response appears to be relatively short-lived, with no significant differences observed between p42 and p30 HPC-7 cells after 12 hours (Fig. 3A-C). How do the authors reconcile this result with the finding that thousands of genes, including inflammatory genes, are downregulated in p30-expressing HPC-7 cells even after 24 hours, as shown in Fig. 1?
18. How many genes from the p30-downregulated set shown in Fig. 1A are differentially regulated by LPS stimulation, both during early and late activation phases? Additionally, how do the p30-upregulated genes in Fig. 1A compare in terms of differential regulation by LPS?
19. How do the authors reconcile the absence of differentially expressed genes (DEGs) between p30 and p42 after 12 hours (Fig. 3) with the significant changes in chemokines observed even after 16 hours (Fig. 4)?
20. Why is CXCL10 absent in Fig. 4C and 4D after LPS stimulation, despite being identified in Fig. 3H as one of the most consistently downregulated genes by p30 following LPS activation?
21. The authors concluded that fos/AP-1 is the main regulator for the inflammatory response. How come that CXCL10 (which they identified among the 30 conserved inflammatory genes in Fig 3H) is not upregulated when FOS is transduced in KO-52 cells in Fig 6H?
22. The authors hypothesized that incomplete activation of inflammatory genes in p30-expressing cells might mitigate the adverse effects of prolonged LPS exposure. While chronic inflammation is known to impair hematopoietic stem cell (HSC) function, this does not necessarily translate to acute myeloid leukemia (AML). Various studies suggest that inflammation in the bone marrow can promote hematological malignancies. Furthermore, the effect of LPS impairment shown in Fig. 4F is minimal and not clearly discernible due to the unfavorable y-axis break, which complicates proper interpretation. It appears that non-LPS-treated p30-overexpressing HPC-7 cells exhibit a 4-5-fold higher cell count compared to non-LPS-treated p42-transduced HPC-7 cells at day 8. Similarly, LPS treatment results in a 4-5-fold increase in cell count for p30-overexpressing cells compared to p42 cells. The impairment in p42 cells due to LPS is minimal compared to the advantage conferred by p30, regardless of LPS stimulation (Fig. 4F). Given the unfavorable y-axis break in Fig. 4F—specifically between p42 untreated and p42 LPS conditions—the authors should remove this break and present the data on a full linear scale to facilitate accurate interpretation. Overall, these results do not strongly support the notion that inflammation is the primary driver behind the competitive advantage of p30 over p42. Additional cell models, ideally primary samples, would be needed to support their hypothesis more convincingly.

Version 1:

Reviewer comments:

Reviewer #1

(Remarks to the Author)

The authors did an great job to revise the manuscript and to address the reviewers' comments. I am fully satisfied with the revision.

Reviewer #2

(Remarks to the Author)

I thank the Authors for the substantive amount of work performed in response to my comments, which I consider to have been sufficiently addressed.

If editorially possible, I would recommend bringing the summary diagram currently in Supplementary Fig. 8 into the main Figures. I also suggest including some of the insightful discussion by the Authors of the Minor Point 2 I raised, as it contextualises their findings in respect of time-dependent cellular evolution of CEBPA mutant leukaemia.

Reviewer #3

(Remarks to the Author)

The authors have conducted a series of new experiments, performed re-analyses, and made significant improvements to their manuscript. They have addressed many of the previous concerns/questions.

The remaining points are as follows: the authors decided to remove the initial Venn diagram and the associated figures related to points 14/15. However, it remains important that the authors discuss the lack of a conserved inflammatory response across the systems they used. Similarly, points 17, 19, and 20 could be briefly included in the discussion section.

The authors did not explicitly mention where each of the reviewer figures has been incorporated into the manuscript or the reasons for not including them. It would be helpful for the reader if these figures were included as supplementary material.

Aside from these points, I have no further comments or questions.

Mutant CEBPA promotes tolerance to inflammatory stress through deficient AP-1 activation

Overall response to reviewers

We thank the reviewers for their thoughtful and constructive comments. We have made a major revision to our manuscript in response to them, and we believe that it has greatly improved as a result. The reviewers raised two main points:

The first is about the use of an AML cellular model to validate our findings in murine cells (Reviewer #2). To address this, we have generated KO-52 cells (AML cells with biallelic CEBPA mutation) expressing wild-type p42 and performed methylcellulose self-renewal assays in the presence of LPS and tunicamycin. These experiments confirmed the same trends observed in mouse cells: p30-expressing cells exhibit increased fitness under LPS challenge, but are less resistant to tunicamycin-induced ER stress. We have also over-expressed Fos and show a similar trend as expressing p42, consistent with the idea that p42-mediated inflammatory gene expression is at least partially driven by Fos. To further elucidate the molecular underpinnings of these findings, we generated new ATAC-seq data. Lastly, we have also re-analysed public Hi-C data from human AML samples to show that loss of regulatory genomic interactions in CEBPA-mutant patients may underlie the reduced Fos expression in AML.

The second main concern is the use of tamoxifen and the expression of fusion constructs. First, we have added the controls requested by Reviewer #1, which consist of cellular fractionation assays to verify the nuclear translocation of fusion proteins. The main concern of Reviewer #3 was that the use of tamoxifen to induce CEBPA-ERT2 nuclear translocation could be interfering with gene expression changes. To address this, we have performed a new RNA-seq in HPC-7 cells expressing the constitutive forms of p30 and p42, without any fusion and without adding tamoxifen. In addition, we have re-analysed RNA-seq data from a previous study that used primary GMPs expressing endogenous p30 and p42. In both cases, the differences in inflammatory gene expression persisted, indicating that they stem from p30 and p42 themselves rather than from tamoxifen treatment.

In addition to these two main points, we have also made new experiments to address the rest of comments, including:

- New CEBPA-ATF4 co-immunoprecipitations using three different CEBPA mutants to gain new mechanistic insights into the CEBPA-ATF4 interaction, as requested by Reviewer #1.
- We have performed ATF4 ChIP-seq in KO-52 cells, to examine the chromatin changes induced in the ER-stress response in AML.
- We have performed a new CEBPA ChIP-seq in HPC-7 cells using CEBPA antibody, to compare it to the ER antibody ChIP-seq and increase experimental robustness.
- ATF4 knockdown experiments to demonstrate a specific functional cooperation with p42, as requested by Reviewer #1.
- We have conducted additional RNA-seqs in KO-52 cells with p42 overexpression to include controls requested by Reviewer #3. More specifically:
 - We have repeated the p42 and EV RNA-seq using spike-in controls for normalization.
 - We have generated p30-overexpressing KO-52 cells as a control instead of empty-vector-transduced cells and performed the RNA-seq with spike-in controls.

Finally, we have expanded the statistical details as requested by the journal. Changes in the manuscript are indicated in red.

A detailed point by point response to the reviewers' questions follows below.

Reviewer #1 (Remarks to the Author):

Cadefau-Fabregat et al. investigates the impact of the CEBPA mutant form p30 in acute myeloid leukemia. They demonstrate the first time that expression of the p30 mutant AML cells have impaired inflammatory gene expression and altered response to LPS. They provide substantial mechanistic insights, by demonstrating the importance of the AP-1 transcription factor family as critical downstream targets of CEBPA in AML. The study is highly comprehensive, with many different models and methods, and offers important novel insights into the role of the CEBPA mutant in AML. The work is overall well done and solid. However, I am not fully convinced about the interplay between CEBPA and ATF4.

We thank the reviewer for the overall positive assessment of our study.

Major points

- 1) The authors used first a rather artificial system via tamoxifen induction of ER-fusion

proteins. This approach can sometimes suffer from leakiness. As control, the authors should demonstrate that both proteins translocate from the cytoplasm into the nucleus and bind to chromatin upon 4-OHT induction.

We thank the reviewer for bringing this to our attention. To address it, we have now performed cellular fractionation of the fusion proteins followed by Western blot. We show that both p42-ERT2 and p30-ERT2 are efficiently translocated from the cytoplasm into the nucleus after 4-OHT treatment (**Supplementary Fig. 1b**). As the reviewer mentions, however, there is a certain degree of leaky translocation already before 4-OHT addition, which is not uncommon in these systems. Our rationale to use an inducible system was to minimize possible premature effects of p30/p42 on chromatin, which could potentially drive premature differentiation of the HPC-7 cells, adding additional elements that could complicate the interpretation of the results. Therefore, to address if the leakiness of the two fusion proteins was driving premature differentiation of HPC-7 cells, we stained cells with differentiation markers prior to 4-OHT treatment. This experiment showed that neither fusion protein increased the expression of differentiation markers (**Supplementary Fig. 1c and Reviewer Fig. 1a**), indicating that the low levels of translocated fusion protein before 4-OHT did not have a significant functional impact in terms of differentiation.

To address if the fusion proteins can efficiently bind chromatin, we isolated the chromatin fraction out of the nuclear fraction and performed Western blot, which shows that both proteins can bind chromatin (**Reviewer Fig. 1b**). This is further supported by the identification of thousands of peaks by ChIP-seq, which we have performed using antibodies against CEBPA and against ER. Notably, the ChIP-seq signals obtained in both experiments are highly enriched at peaks previously identified in published ChIP-seq data for p30 and p42 in GMPs expressing the endogenous proteins (Jakobsen et al. Sci Adv 2019 PMID: 31309149), thereby validating the binding profiles generated in this study (**Supplementary Fig. 5b, Reviewer Fig. 1c**). In summary, we are very confident that both fusion proteins bind chromatin and show a binding pattern highly similar to the endogenous proteins.

Reviewer Fig. 1. a) Flow cytometry histograms of the granulocytic marker GR-1 and the monocyte/macrophage marker CD11b of HPC-7 cells overexpressing p42, p30 or an empty vector before treatment with 4-OHT; b) Western blot against ER of the chromatin fraction of HPC-7 p42 and p30 cells after treatment with 4-OHT; c) α -CEBPA and α -ER ChIP-seq signal enrichment at publicly available p42/p30 peaks from GMPs (Jakobsen et al. PMID: 31309149). Left: HPC-7 p42-ERT2 cells. Right: HPC-7 p30-ERT2 cells.

2) There is no explanation provided why p42, but not p30, is interacting with ATF4. Both proteins possess the dimerizing bZIP domain and it appears illogical why p30 cannot interact with ATF4. AlphaFold 3 predicts an interaction for both p30 and p42 with ATF4. Thus, some more mechanistic insights would strengthen the manuscript.

The reviewer raises an important point, and we agree that additional experiments will improve the manuscript. bZIP proteins represent one of the largest families of transcription factors (Lambert et al., *Cell* 2018, PMID: 29425488) and, as obligate dimers, form a highly selective interactome despite the bZIP's highly homologous sequence (Newman & Keating, *Science* 2003, PMID: 12805554; Rodriguez-Martinez et al., *eLife* 2017, PMID: 28186491; Bendel et al., *Nat Commun* 2024, PMID: 39402041). In the specific case of CEBPA, it has been previously demonstrated that the p30 and p42 isoforms have isoform-specific interactors, including bZIP proteins such as Jun, FosL2, Atf3 or Ddit3 (Grebien et al., *Nat Chem Biol* 2015, PMID: 26167872; Ramberger et al., *iScience* 2021, PMID: 34189442). However, since these experiments were conducted in other cell types and under different conditions, the differential

interaction with ATF4 was not identified. Nevertheless, our finding that a bZIP protein (ATF4) differentially interacts with p30 and p42 is consistent with these previous studies.

One approach to gain insights into the dimerization properties of these proteins would have been to generate and isolate the recombinant proteins for *in vitro* interaction studies. However, despite our efforts, the p30 isoform was predominantly insoluble, and we were unable to achieve sufficient solubilization. Consequently, we reasoned that additional mechanistic insights could be obtained from over-expressing modified versions of CEBPA to perform co-immunoprecipitation assays. Specifically, we hypothesized that regions outside of the bZIP domain might influence dimerization capacity.

The bZIP domain is located in the C-terminal region of the protein, and the majority of the N-terminal region constitutes an intrinsically disordered region (IDR), therefore lacking a well-defined three-dimensional structure (**Reviewer Fig. 2a**). In a recently published study, we demonstrated that this IDR provides CEBPA with the capacity to undergo phase separation, enabling the spontaneous formation of biomolecular condensates through weak interactions (Christou-Kent, Cuartero, Garcia-Cabau et al., *Cell Reports* 2023, PMID: 37516962). We proposed that this phase separation ability facilitates the formation long-range genomic interaction hubs which enable the coordinated activation of transcriptional programs by CEBPA. Interestingly, regularly spaced aromatic residues within IDRs have been shown to promote phase separation in numerous transcription factors, including CEBPA (Naderi et al., *Nature Cell Biology* 2024, PMID: 38969762). Since the IDR represents the primary structural difference between p42 and p30, we hypothesized that the phase separation capacity conferred by this region may also influence interactions with other proteins, specifically with ATF4.

To test this, we expressed a modified version of CEBPA in which all aromatic residues were mutated to other types (**Reviewer Fig. 2b**). This mutant, termed AroLite, displays impaired phase separation (Christou-Kent, Cuartero, Garcia-Cabau et al., *Cell Reports* 2023, PMID: 37516962). We performed co-immunoprecipitation (co-IP) experiments and probed for ATF4. As shown in the revised **Supplementary Fig. 7b**, the AroLite mutant exhibited a markedly reduced interaction with ATF4 compared to the wild-type (WT) protein. Conversely, we expressed another modified version of CEBPA in which the aromatic residues were evenly dispersed with perfectly uniform spacing (AroPERFECT, **Reviewer Fig. 2b**) and that enhances condensate formation properties (Naderi et al., *Nature Cell Biology* 2024, PMID: 38969762). In this case, the interaction between CEBPA AroPERFECT and ATF4 was noticeably stronger than that of the WT protein (**Supplementary Fig. 7b**). This observation is

consistent with public ChIP-seq data showing that the AroPERFECT mutant binds approximately 10,000 additional peaks compared to WT CEBPA, with these additional peaks being less enriched for the CEBPA motif and more enriched for motifs of other bZIP transcription factors (Naderi et al., *Nature Cell Biology* 2024, PMID: 38969762). Overall, these results indicate that the presence and distribution of aromatic residues within the IDR are critical for interaction with ATF4 and suggest the interesting possibility that CEBPA's phase separation capacity plays an important role in mediating this interaction.

Early studies on CEBPA domains identified three trans-activating elements: TE-I, TE-II, and TE-III (**Reviewer Fig. 2a**, Nerlov & Ziff, *Genes & Dev* 1994, PMID: 8314088). While these elements are located within the IDR and therefore lack a conserved structure, functional studies have demonstrated their gene-activating capacity in transgene assays. TE-I and TE-II are specific to p42, whereas TE-III is shared by both isoforms and mediates interaction with the SWI/SNF nucleosome remodeling complex (Nerlov, *Nat Rev Cancer* 2004, PMID: 15122210). Given that p42 is the only isoform capable of interacting with ATF4, we hypothesized that TE-I and TE-II might be necessary for this interaction, while TE-III would be dispensable, potentially explaining why p30 does not interact with ATF4. To test this hypothesis, we expressed a mutant version of p42 lacking the TE-III element (CEBPA Δ TE, **Reviewer Fig. 2b**) and performed co-IP with ATF4 (**Supplementary Fig. 7b**). Interestingly, the mutant lacking TE-III lost its ability to interact with ATF4, indicating that this domain is essential for the interaction in p42, despite being present in p30, which does not interact with ATF4. This result suggests two possible interpretations: either TE-III itself is directly required for the interaction with ATF4, or the deletion of the aromatic residues contained within this region disrupts the interaction similarly to the AroLite mutant.

Overall, these findings demonstrate that the dispersion of aromatic residues within the IDR is critical for interacting with ATF4 and that TE-III is essential for this interaction. This underscores the importance of protein sequences outside the bZIP domain in regulating interactions with other proteins and suggests a potential role for phase separation in the selectivity of bZIP protein dimerization. While these results are promising, further investigation would require substantial additional work, which falls beyond the scope of this study. We plan to follow up on this by exploring the role of CEBPA phase separation in mediating interactions with ATF4 and potentially with other bZIP proteins in future studies.

Reviewer Fig. 2. a) Schematic of the CEBPA protein, indicating the first aminoacids of p42 and p30; the three transactivation domains (TE-I, TE-II and TE-III); the location of aromatic residues; and the disorder prediction score. Figure adapted from Christou-Kent et al. *Cell Rep* 2023 PMID: 37516962. **b)** Schematic representation of the three mutants tested. The AroLITE mutant lacks all aromatic residues in the IDR, whereas the AroPERFECT has regularly spaced aromatic residues. The Δ TE-III mutant lacks the TE-III domain, which is shared by p42 and p30. Figure adapted from Naderi et al. *Nat Cell Biol* 2024 PMID: 38969762.

3) The authors propose that p42 particularly cooperates with ATF4 for gene regulation. To make this statement stronger, I would recommend deleting ATF4 from the HPC-7 cells. One would expect weaker differences upon tamoxifen induction of p42 versus p30. This could be assessed by RT-qPCR of key dysregulated genes from Fig. 1A.

We thank the reviewer for proposing this experiment, which we have now performed and included in the revised manuscript. We used shRNAs to knock down *Atf4* in HPC-7 cells expressing either p42-ERT2 or p30-ERT2, achieving a reduction of approximately 40–60% in transcript levels (**Supplementary Fig. 7d**). Nuclear translocation of p30 and p42 was then induced using 4-OHT treatment, followed by qPCR analysis of a panel of eight genes identified as deregulated in Fig. 1A. These genes were selected based on the presence of both ATF4 and CEBPA binding sites linked by HiChIP interactions, indicating that they are likely co-regulated by both factors. As expected, all eight genes showed reduced expression in p30-expressing cells compared to p42-expressing cells when transduced with a control plasmid expressing a scrambled shRNA. However, upon knockdown of *Atf4*, the differences between p42 and p30 became weaker (*Lyz2*, *Gadd45a*, *Serpinb1a*, and *Cd14*) or were even reversed (*Fos*, *Zeb2*, *Ccl4*, and *Gadd34*) (**Supplementary Fig. 7e**). These findings confirm the reviewer's prediction that ATF4 knockdown reduces the expression differences between p42 and p30-expressing cells, and is consistent with our overall findings, which suggest a specific cooperative role between p42 and ATF4.

Minor Points:

1) line 378: p42-expressing > p42-expressing

Now corrected.

2) lines 455-458: Where is this result shown?

This upregulation is presented in Fig. 2i, and we have now included a reference to this figure in the main text.

3) Fig. 7B should also include the ATF4 ChIP-Seq signal.

We have now added a heatmap of ATF4 ChIP-seq signal next to the H3K27ac signal.

4) it would be nice to present a figure, that summarizes the main points of the manuscript

We thank the reviewer for the suggestion. We have designed a model summarizing how CEBPA isoforms differently affect FOS and ATF4 function (**Supplementary Fig. 8**).

5) The title should make clearer that the results refer to AML. No data are shown that the conclusion is more general, beyond AML.

We appreciate the reviewer's concern regarding the need to clarify that our conclusions do not apply to other cancers or cellular lineages. Initially, we did not consider it necessary to include "AML" in the title, as CEBPA mutations are only found in AML, and therefore results were unlikely to be extrapolated to other cancers. Additionally, we chose not to add "AML" to the title because many of our experiments were not performed in strictly AML cells, such as the models of p30 expression in HSPCs and macrophages. Furthermore, another reviewer raised concerns about the relevance of our findings to AML, which we are addressing in this revision. We believe that maintaining the current title strikes a balance, as "Mutant CEBPA" is inherently linked to AML. Nonetheless, to further address the reviewer's concern, we have added a sentence in the discussion explicitly stating that our results are not applicable to other cancers (lines 743-744). We hope the reviewer understands our reasoning and finds this approach satisfactory.

6) The Fig. legends for Fig. 4D should make clearer that the shown data are mass-spectrometry data.

We have now specified in the figure legends of panels 4b, c and d and Supplementary Fig. 4a that it is mass spectrometry data.

7) The authors used publicly available ATF4 ChIP-seq data, but it is not fully clear how the

data were used. I would recommend to explain this point better. Also why did the authors use two different sets of ATF4 ChIP-Seq data, and not just the better one?

The question we aimed to address was whether genes bound by ATF4 (either directly at the promoter or through looping to other ATF4 binding sites) were more likely to be differentially expressed between p42- and p30-expressing cells. To investigate this, we searched publicly available datasets in hematopoietic cell types. As we were unable to find suitable datasets with sufficient quality for HSCs or myeloid progenitors, we initially used ChIP-seq data from total bone marrow cells (GSE132681). However, when we overlapped these data with our H3K27ac peaks, only 64 peaks remained, which was insufficient for statistically robust conclusions. To improve the analysis, we supplemented this dataset with ChIP-seq peaks from an experiment in bone marrow-derived dendritic cells (GSM881130), which was the closest available alternative. This dataset was of higher quality and yielded 1,300 peaks overlapping with our H3K27ac peaks. We then merged these with the 64 peaks from the total bone marrow dataset. Our rationale for merging the two datasets was to reduce bias and strengthen the robustness of the results. However, in line with the reviewer's suggestion, we have now repeated the analysis using only the second dataset (GSM881130), as the first dataset provided minimal additional information. The results are essentially the same and, therefore, to enhance the clarity of the paper, we have substituted this analysis in Fig. 7b-c, 7e-i and updated the description in the revised manuscript.

Reviewer #2 (Remarks to the Author):

In this manuscript, Cadefau-Fabregat et al, investigate the differential transcriptional and gene regulatory consequences of canonical p42 and AML-associated p30 isoforms of CEBPA. Whilst DNA binding and transcriptional activation are broadly unchanged, the authors identify a selective defect in activation of AP1-dependent inflammatory and early-response genes. Through analysis of HPC-7 cells with inducible expression of p30, complemented by conditional *Cebpa^{Fl}/p30*; R26-Cre-ER BM progenitors and primary macrophages, and p42 rescue of KO-52 cells, the authors show that unique expression of the p30 isoform affects acute inflammatory gene expression including molecular and cellular responsiveness to LPS by interfering with the strength of enhancer activation at regulatory regions also bound by AP-1 family members. The weaker and delayed inflammatory effects are partially mediated by Fos downregulation and may reduce pro-apoptotic effects of the inflammatory process, although this is not evident in primary cells. The authors suggest that the relative gain in fitness of p30-expressing cells upon prolonged inflammation may facilitate the malignant process, but there is limited or no data to support this assertion beyond the unchanged apoptosis and higher expansion of HPC-7 p30+ cultures in the presence of LPS and unchanged colony formation

by p30 BM progenitors. In a related, but separate mechanistic exploration, the authors also show a specific effect of p30 in ATF4-mediated stress responses, which they suggest may be explored as a therapeutic mechanism. Specifically, p30, unlike p42, does not interact with ATF4 (by co-IP), resulting in reduced transcriptional activation at CEBPA/ATF4 sites. In this case, ER stress leads to increased apoptosis in less responsive p30-expressing HPC-7 cells, highlighting a potential drug vulnerability of CEBPA mutant AML cells.

The study is well designed and well executed and the conclusions as to the differential transcriptional regulatory consequences of p42 and p30 isoforms of CEBPA are justified within the context of the cell models used. However, it is not clear to me which of these mechanisms apply to the leukemia process, as all the associations with AML patient samples are correlative on gene expression profiles that may to a large extent reflect cell composition, with no functional validation. Given the centrality of p30 expression to CEBPA mutant leukemia, this should be addressed directly to support the wider implications of the data. HPC-7 cells or differentiated macrophages are not representative of CEBPA mutant leukemia cells.

We thank the reviewer for the overall positive assessment of our study and for the constructive comments. We agree with his general comment and we have made additional experiments with cellular models of AML to further support our interpretation of the findings.

MAIN POINTS

1. The Authors should investigate the contribution of diminished AP-1 driven inflammatory gene transcription to CEBPA-mutant leukemia by:

a. determining the impact of LPS treatment on p30-transformed cells, either in vitro through serial replating, or, if feasible, in vivo through leukemia latency in the presence of a chronic inflammatory stimulus;

conversely, the role of FOS down-regulation in dampening inflammatory response and facilitating transformation can be assessed through up-regulation of Fos.

We thank the reviewer for suggesting these experiments. To investigate the role of p30 in the inflammatory response in an AML model, we have chosen the KO-52 cell line, which, to date, is the only established AML cell line with biallelic CEBPA mutations (Heyes et al., *Leukemia* 2021, PMID: 33623142), and we had already conducted transcriptomic experiments demonstrating that p42 re-expression partially rescues inflammatory gene expression (**Fig. 2**).

Furthermore, these cells exhibit reduced FOS expression compared to cells with p42 re-expression (**Fig. 2**). For these reasons, we believe that KO-52 represents an optimal model to address the reviewer's concerns.

First, we assessed the colony-forming capacity of KO-52 cells overexpressing either wild-type p42 (which they do not endogenously express) or p30 (as a control) in methylcellulose. These cells were plated in methylcellulose supplemented with LPS, and colonies were counted 14 days later. While LPS only minimally reduced the number of colonies formed by p30-expressing cells, it significantly reduced the number of colonies formed by p42-expressing cells (**Supplementary Fig. 4e**). This finding is consistent with the reduced colony formation observed in mouse primary progenitors (**Fig. 4e**) and the reduced cell numbers in HPC-7 cells exposed to LPS (**Fig. 4g**), indicating that these observations are recapitulated in an AML cellular model.

As suggested by the reviewer, we then assessed the contribution of AP-1-driven inflammatory gene transcription to CEBPA-mutant leukemia by upregulating FOS expression and evaluating cellular fitness in response to inflammatory signals. To do this, we stably transduced KO-52 cells with FOS-expressing plasmids (**Supplementary Fig. 6c**), plated them in methylcellulose supplemented with LPS, and colonies were counted after 14 days. LPS treatment caused a more pronounced average reduction in the number of colonies formed by FOS-expressing cells compared to control cells, although the p-value of the t-test was not below 0.05 due to variation in colony number counts. We then replated these cells for a second round and observed that control cells showed an increase in colony numbers under LPS treatment compared to FOS-expressing cells (**Supplementary Fig. 6k-l, Reviewer Fig. 3**). Since we noticed that LPS reduced both the colony number and the total cells in the initial plating, we calculated the cumulative CFU potential, which accounts for the initial expansion difference in the overall clonogenic activity (Higa et al., *J Exp Med* 2021, PMID: 33914855). This showed that while control cells showed minimal reduction when exposed to LPS, FOS-expressing cells presented a significantly reduced CFU potential (**Supplementary Fig. 6m-n**). We attempted a third replating, but due to technical issues, only one replicate grew successfully. For this reason, this result is not included in the revised manuscript, but it is presented in **Reviewer Fig. 3** and demonstrates a consistent trend, with LPS having a greater negative impact on FOS-expressing cells compared to controls. Overall, these experiments indicate that the partial rescue of inflammatory gene expression by FOS leads to a greater fitness reduction in FOS-expressing cells under inflammatory conditions. This suggests that reduced FOS expression confers a fitness advantage to CEBPA-mutant cells in response to inflammation.

While we acknowledge that *in vivo* transplantation assays would further strengthen our conclusions and we plan to conduct them in the future, we were unable to include them within the timeframe of this revision and the scope of this paper. This study primarily focuses on the molecular mechanisms underlying our findings, which we believe lays the groundwork for future research and opens up various avenues for potential *in vivo* experiments that will be addressed in subsequent studies.

Reviewer Fig. 3. Self-renewal capacity of KO-52 FOS. Ratio of the number of colonies grown in methylcellulose supplemented with LPS or untreated in KO-52 transduced with an empty vector (EV) or FOS during three CFU platings. Shown are two different ways to represent the results, in box plots and line plot to appreciate the tendency over time.

b. inspecting p30, AP-1 gene family and H3K27ac chromatin binding at inflammatory and control loci in vitro (or in vivo)-transformed cells.

To investigate the chromatin state in transformed cells at the molecular level, we performed ATAC-seq in p42- and p30-transduced KO-52 cells. We identified 11,674 peaks with increased accessibility in p42 cells (adj.P<0.05, **Supplementary Fig. 6c**), and motif enrichment analysis of these peaks resembled the enriched motifs among p42-induced H3K27ac peaks in HPC-7 cells shown in **Fig. 5h**, with the CEBP:AP1 motif among the top five most significantly enriched (**Supplementary Fig. 6d**). To determine whether FOS was one of the AP-1 factors binding these sites, we plotted the fold-change in accessibility across all sites, revealing that FOS-bound sites were associated with higher ATAC signal (**Supplementary Fig. 6e**). Next, we identified genes displaying increased promoter accessibility, separating those with (N = 154) and without (N = 394) FOS binding. Accordingly, the genes bound by FOS were more strongly upregulated in p42 cells (**Supplementary Fig. 6f**). GO analysis showed that FOS-bound genes were enriched for inflammatory terms, suggesting a role for FOS in enhancing accessibility at inflammatory promoters (**Supplementary Fig. 6g**). Finally, we also analysed CEBPA and FOS binding patterns in KO-52 cells. We found that inflammatory genes co-bound

by CEBPA and FOS exhibited a greater expression increase in p42-transduced cells compared to p30-transduced cells than did CEBPA-FOS co-bound house-keeping genes lacking an inflammatory function or inflammatory genes not bound by these factors (**Supplementary Fig. 6h**).

In addition to the analyses performed in KO-52 cells, we examined the chromatin state of primary AML samples. Specifically, we re-analyzed recently published Hi-C datasets from AML patients with CEBPA mutations (n=6) and compared them to AML samples with non-CEBPA mutations (n=16) (Xu et al., *Nature* 2022, PMID: 36289338). While we also attempted to re-analyse ATAC-seq data from the same study, the limited number of CEBPA-mutant samples (n=2) prevented us from drawing meaningful conclusions. Using the Hi-C data, we performed a virtual 4C analysis centered on the *FOS* gene. This analysis revealed that interactions between *FOS* and its surrounding regulatory elements were reduced in CEBPA-mutant AML samples compared to other AML types (**Supplementary Fig. 6b**). These findings align with the reduced interactions observed at the *Fos* locus in mouse HPC-7 cells (**Fig. 6f**) and suggest that diminished chromatin interactions at the *FOS* locus in AML may contribute to its reduced expression.

c. measuring p30 and ATF4 chromatin binding at stress loci and measuring responsiveness of p30-transformed cells, in vitro or in vivo, to ER stress.

To address this question, we first assessed the viability of KO-52 cells overexpressing either p42 or p30 in response to tunicamycin, which induces an ATF4-dependent ER stress response. As shown in **Supplementary Fig. 7f**, a 5-day tunicamycin treatment triggered cell death in both p30- and p42-overexpressing KO-52 cells, but cell death was significantly more pronounced in p30-expressing cells, indicating a protective effect of p42 expression under acute ER stress. Next, we examined the impact of ER stress on the self-renewal capacity of KO-52 cells in methylcellulose culture. As expected, colony formation was markedly reduced when methylcellulose was supplemented with tunicamycin. Consistent with the cell survival analysis, p42-expressing cells formed significantly more colonies than p30-expressing cells (**Supplementary Fig. 7g**). These findings align with the results in HPC-7 (**Fig. 7m-n**) and collectively indicate that cells lacking wild-type p42 are more susceptible to ER stress-induced cell death and fitness loss.

To investigate the molecular mechanisms underlying this phenotype in KO-52 cells, we first performed ATF4 ChIP-seq in this cell line, yielding 16,470 peaks, and found that genes bound by ATF4 at their promoter or gene body were upregulated in p42-rescued KO-52 cells but not

in p30-expressing cells (**Supplementary Fig. 7h**). We then conducted ATAC-seq on p30- and p42-expressing KO-52 cells under steady-state conditions and after 8 hours of tunicamycin treatment. At baseline, ATF4 binding sites were significantly more accessible in p42-expressing cells compared to p30-expressing cells (**Supplementary Fig. 7i**). Following tunicamycin treatment, chromatin accessibility increased in both cell lines, but p42-expressing cells maintained higher accessibility at ATF4 sites than p30-expressing cells (**Supplementary Fig. 7i**). We then identified ATAC-seq peaks that either increased or decreased following tunicamycin treatment in p42-expressing cells. In both cases, the magnitude of these changes was greater in p42-expressing cells than in p30-expressing cells, where the stress response triggered milder changes, albeit in the same direction (**Supplementary Fig. 7j**). Together, these results suggest that p42 expression enhances accessibility at ATF4 binding sites and may explain the differential expression of ATF4-target genes in CEBPA-mutant AML shown in **Fig. 7j-l**.

2. To consolidate the co-regulation of p30-target loci on AP1 factors, the Authors should perform ChIP-seq analysis of AP1 family, including ATF4. binding at p30 target loci with differential transcription.

To further investigate the co-regulatory dynamics between CEBPA isoforms and AP-1 factors, we have focused our analysis on the two AP-1 factors highlighted in the manuscript: FOS and ATF4. For FOS, we re-analysed publicly available mouse FOS ChIP-seq data (from ChIP-Atlas <http://dx.doi.org/10.15252/embr.201846255>). We first identified 675 genes that displayed p42-specific HiChIP interactions with FOS binding sites (using our H3K27Ac HiChIP data), and we found that these genes were on average downregulated when we compared gene expression in p30- versus p42-expressing HPC-7 cells (**Reviewer Fig. 4a**). Of these, 310 genes were significantly downregulated by DESeq2 analysis (adj.P < 0.05). These genes are also downregulated in AML samples with N-terminal CEBPA mutations but not those with in C-terminal mutations, and their expression is rescued by p42 over-expression in KO-52 cells (**Reviewer Fig. 4b**). We further filtered for genes downregulated in both HPC-7 and AML with N-terminal mutations and upregulated in KO-52 p42, yielding a core set of 141 likely p42–FOS co-regulated genes. A GO analysis of these genes reveals that inflammatory response is the most enriched term (**Reviewer Fig. 4c**). Overall, these analyses indicate that a large fraction of FOS-bound genes in p42-expressing cells contribute to the increase in inflammatory gene expression compared to p30 cells. Finally, as explained above (point 1b) we also analysed CEBPA and FOS binding patterns in KO-52 cells, revealing inflammatory genes co-bound by CEBPA and FOS exhibited a greater expression increase in p42-transduced cells compared to p30-transduced cells than did CEBPA-FOS co-bound house-keeping genes lacking an

inflammatory function or inflammatory genes not bound by these factors (**Supplementary Fig. 6h**). We have incorporated the KO-52 data into the manuscript, while the corresponding HPC-7 findings appear in the reviewer figure 4, as we considered them potentially redundant.

To study the dynamics of co-binding of CEBPA and ATF4 and their effect on gene expression, we first divided genes into those that showed exclusive binding of p42 and those with exclusive binding of p30 (excluding genes with binding of both) in HPC-7 cells. We then further divided these two groups into those with and without ATF4 binding. Finally, we plotted the fold-change in expression comparing p30- versus p42-HPC-7 cells, which shows that the genes with specific p42 and ATF4 co-binding are on average more downregulated than the rest three groups (**Reviewer Fig. 4d**), in line with the results shown in Fig. 7e-l and indicating a cooperative role between p42 and ATF4.

Reviewer Fig. 4. a) Gene expression changes (log₂) in p30- versus p42-expressing cells of the 675 genes that displayed p42-specific HiChIP interactions with FOS binding sites. b) GSEA plots of the 310 downregulated genes (adj.P < 0.05) in AML samples with C-terminal CEBPA mutations, N-terminal mutations, and p42 over-expressing KO-52 cells. c) Manhattan plot of the GO analysis of the 141 genes downregulated in both HPC-7 and AML with N-terminal mutations and upregulated in KO-52 p42, with a table of the top 10 GO terms. d) Fold-change in expression (log₂) comparing p30- versus p42-HPC-7 cells of genes with specific p42 and ATF4 co-binding.

MINOR POINTS

1. The Authors should clarify the significance of inducing p30 expression in differentiated macrophages to study the impact of the respective isoform on inflammatory gene expression.

What is their abundance and relative inflammatory gene expression in CEBPA mutant leukemia?

We apologize for not being clearer in the manuscript regarding the rationale behind using macrophages, which has now been revised for increased clarity (lines 139-148). Our reasoning was as follows: since the expression of p30 and p42 in progenitors could potentially skew the myeloid differentiation trajectory, the observed effects on inflammation might simply result from differences in the composition of progenitor populations or variations in the pace of differentiation. These factors could, in turn, explain the differences in inflammatory gene expression observed in bulk RNA-seq analyses. To address this concern, we designed the experiment with macrophages to allow cells to follow their normal differentiation trajectory without interference. Specifically, we added 4-OHT (to delete the p42 WT allele) only at the last stages of differentiation. This ensured that we were comparing equivalent populations of fully differentiated macrophages, without differences in cell maturity (**Supplementary Fig. 1p**). Indeed, our data showed that changes in inflammatory gene expression were evident even when comparing macrophages of the two genotypes (**Fig. 1e,f**), confirming that the observed differences were not due to variations in the pace or state of cellular maturation.

Besides this clarification, we agree with the reviewer that investigating whether macrophages play a role in CEBPA-mutant leukemia would be very interesting. This concept aligns with studies showing that altered inflammatory gene expression in Tet2-mutant leukemia mainly arises in mutant monocytes (Yeaton et al. *Cancer Discov* 2022 PMID: 35924979, Heimlich et al. *Blood Adv* 2024 PMID: 38507736). We recognize the importance of this point and plan to explore it in future studies.

2. How do the Authors position the defective AP-1-mediated inflammatory and ER stress responses in terms of their contribution to CEBPA AML? Is there cell type specificity, and how does a possible cell specificity CEBPA mutant leukemia mice compare to WT? Or do the Authors think that they may have a differential contribution at different stages in leukemia initiation and progression?

While our data does not allow us to specifically answer these questions, we propose two possible scenarios in which resistance to inflammatory signals could play a pro-leukemic role. In the first scenario, protection from inflammation would be critical during the early stages of leukemia initiation, or even in pre-leukemic stages. Here, acute inflammatory episodes or chronic inflammation could act as a strong selective pressure, favoring the expansion of inflammation-resistant clones. In the second scenario, resistance to inflammation becomes

more important at later stages, when the leukemic microenvironment is characterized by a highly altered pro-inflammatory milieu (Chen et al., *Blood Cancer Discov* 2023, PMID: 37470778; Schepers et al., *Cell Stem Cell* 2015, PMID: 25748932). In this context, defective inflammatory responses might allow CEBPA-mutant cells to evade inflammatory damage or stress, conferring a selective advantage. Distinguishing between these two scenarios is experimentally challenging. However, emerging strategies to mimic the sequential acquisition of mutations in mouse models (Bowman et al., *Cancer Cell* 2024, PMID: 39532065) may be good experimental models to address these questions. Additionally, investigating mutation-specific interactions within the leukemia-bone marrow niche will be equally essential to understand the role of extracellular signals and stress on clonal fitness and evolution.

The situation may be slightly different when considering the differential response to ATF4-mediated stress, which may potentially extend beyond ER stress to include other stresses within the integrated stress response (ISR). Importantly, it has been demonstrated that HSCs are more susceptible to ER and ISR-mediated cell death than committed progenitor cells, likely as a mechanism to preserve the integrity of the HSC pool (Van Galen et al. *Nature* 2014, PMID: 24776803, Van Galen et al. *Cell Rep* 2018 PMID: 30380403). Additionally, recent single-cell RNA-seq analyses of AML patient samples have revealed that the effects of CEBPA mutations can vary depending on the maturation stage of the blasts, including differences in the expression of AP-1 members (Adamo et al. *Leukemia* 2023 PMID: 36333583). Based on this, we hypothesize that within the CEBPA mutant blast population, increased sensitivity to ER stress may be particularly relevant to immature blasts and leukemic stem cells (LSCs) rather than to more differentiated blasts. To explore this hypothesis, we are planning follow-up studies using single-cell technologies to distinguish subpopulations and maturation stages, aiming to uncover vulnerabilities associated with these stress responses.

3. Can the Authors please clarify the nature of the progenitor colonies quantified in 4E upon LPS treatment of WT and p30 progenitors. This is important to understand the cellular effects of chronic inflammation in the context of p30 unique expression.

We thank the reviewer for this important question. Since CEBPA plays a key role in the development of myeloid progenitors, we used MethoCult M3534, a methylcellulose-based medium that specifically supports the growth of myeloid colonies, mainly granulocyte-macrophage progenitors (CFU-GM). As a result, the colonies observed in Fig. 4e primarily consisted of CFU-GM. Upon visual assessment, we found no significant differences in colony type or frequency between genotypes or treatments. This information has been added to the manuscript (line 326-327).

Reviewer #3 (Remarks to the Author):

The transcription factor CEBPA is frequently mutated in acute myeloid leukemia (AML), resulting in the production of a shorter isoform known as p30. Both the canonical 42-kDa isoform (p42) and the AML-associated p30 isoform bind chromatin and regulate transcription. However, the specific transcriptional programs governed by each isoform and their connection to the selective advantage in AML remain poorly understood. The authors focus on this knowledge gap, particularly the unclear mechanisms by which the differential gene regulation by p30 confers a selective advantage in CEBPA-mutant AML. To address this, they aimed to identify novel transcriptional programs differentially controlled by the CEBPA isoforms and to elucidate the molecular mechanisms driving these distinct transcriptional outcomes. Their findings indicate that p30 fails to promote the normal expression of inflammatory pathways, resulting in a diminished ability to mount an effective inflammatory response to LPS. This deficiency, they conclude, grants p30-expressing cells increased resistance to the harmful effects of prolonged inflammatory signals. Mechanistically, these differences are attributed to the differential regulation of AP-1 family proteins. The authors suggest that these discoveries establish a novel link between mutant CEBPA, inflammation, and the stress response.

To investigate the distinct functions of p42 and p30, the authors utilized tamoxifen-based experimental models. Initially, they used CEBPA transgenes fused to an estrogen receptor binding domain (ERT2) fragment, allowing them to control the nuclear entry of p42 and p30 with 4-hydroxytamoxifen (4-OHT). Through these constructs, they observed differences in inflammatory signaling between p30 and p42. In subsequent experiments, they employed mouse models in which p42 was deleted using Cre-ERT2.

A critical point to consider is that tamoxifen itself modulates various pathways, including the activation of endogenous estrogen receptors (ER) and AP-1 (PMID: 21233487, 29467493). Therefore, it is plausible that the observed transcriptional differences could stem from tamoxifen-activated ER and AP-1 interacting differently with the p42 and p30 isoforms in terms of binding affinities and protein interactions. The experimental design appears to lack the necessary controls to address these possible interactions properly. Notably, it seems that the empty vector was not treated with 4-OHT, and furthermore instead of an empty vector, the proper control should have included the ERT2 fragment without p42 or p30, treated with 4-OHT. Additionally, wild-type (WT) mice do not appear to have been treated with 4-OHT, further indicating that tamoxifen-related effects have not been adequately controlled.

Another concern arises from the initial experiments where the CEBPA isoforms were expressed as fusion proteins with the ERT2 fragment in cell models. The fusion of ERT2 to p42 and p30 could alter their DNA-binding properties and protein interactions, potentially to different extents between the isoforms, that could contribute to the transcriptional changes observed in inflammation-related pathways.

The authors noted that the transcriptional changes associated with p30 and p42 had not previously been linked to inflammation, despite extensive studies on these proteins. This is surprising because significant differences in inflammatory pathways would likely have been easily identified if they were a major factor between the isoforms. This prompts caution in interpreting the results from the specific models used in the study. In summary, the primary concern is that the inferred differences in inflammatory responses between p30 and p42 may be primarily driven by tamoxifen-induced effects. This concern is a recurring theme in the points raised below.

We thank the reviewer for the insightful assessment of our study. We understand the concerns with ERT2 fusion proteins and, to address them, we have made new experiments and analyses. This is all explained in the point-by-point responses below, which show that our results are not due to the fusion systems used. However, we would like to note three general comments:

1. Concerning one of the comments above, we would like to clarify that HPC-7 cells expressing the empty vector control, as well as control primary macrophages and progenitor cells from WT mice, were indeed treated with 4-OHT to account for tamoxifen-related effects. We have further clarified this point in the methods section (lines 792 and 800).
2. We understand the reviewer's concern regarding the fusion of an ERT2 to p42 and p30, but we would like to stress that the primary cells (progenitors and macrophages) do not express CEBPA-ERT2 fusions, but a Cre-ER fusion which is used to delete loxP sites. Thus, in this case there is no direct interaction between CEBPA and ER and therefore it is much more unlikely that any change in inflammatory gene expression can be influenced by that.
3. Regarding the possible effects driven by tamoxifen through the activation of endogenous ER, we find no detectable expression of the ER gene by transcript (RNA-seq) or by protein (by Western blot) in HPC-7 cells, neither before or after tamoxifen treatment and therefore we do not expect to have secondary effects.

Questions/ Points:

1. What are the gene expression changes in HPC-7 cells if the experiments were conducted using constitutively expressed plasmids for p42 and p30 instead of tamoxifen-inducible ones? Specifically, how significantly are inflammatory pathways changed under these conditions?

To address the possibility that altered inflammatory transcription is caused by the ERT2 fusions rather than by p30 or p42 themselves, we stably transduced HPC-7 cells with plasmids constitutively expressing either p42 or p30 (without any fusion) and performed RNA-seq. When comparing p30- and p42-expressing cells, we identified 335 upregulated and 448 downregulated genes ($\text{adj.P} < 0.05$; $\log_2\text{FC} > |1|$), which is similar in magnitude to the number of deregulated genes reported in the inducible system. We first performed GSEA analysis with this new RNA-seq dataset, which confirmed a marked and significant downregulation of inflammatory genes, including the AML-inflammation signature shown in **Fig. 1j** (**Supplementary Fig. 1g**). To directly compare this system with the inducible system, we specifically analysed each one of the top ten downregulated gene ontology sets shown in **Fig. 1b** by performing GSEA in all of them using the results of the new p42 vs p30 comparison. Crucially, these analyses showed that all of them were also downregulated (**Supplementary Fig. 1h**). Next, we analyzed the gene ontology terms enriched in both datasets. We focused on terms enriched among downregulated genes, and observed that those that were significant in both systems ($p\text{-value} < 0.05$) were predominantly related to the inflammatory response (**Supplementary Fig. 1i-j**).

Overall, these data indicate that the downregulation of inflammatory gene expression persists in cells expressing p30 or p42 without ERT2 fusions, suggesting that the effect is driven by the isoforms themselves rather than by any artifact of the fusion proteins. While these controls are important to rule out ERT2 fusion effects, we have included them as Supplementary data because constitutive expression of these isoforms has limitations compared to the inducible system, which remains on the main figure. Specifically, time-controlled expression in the inducible system prevents potential premature differentiation of these cells and minimizes secondary effects of altered transcripts.

2. Supplementary Fig. 1B is missing data for differentiation in control cells. Specifically, what are the expression levels of GR1 and CD11b in HPC7 cells with the control vector, both with and without tamoxifen induction?

We thank the reviewer for pointing this out. The expression of GR1 and CD11b in cells transduced with the control vector remains unchanged between untreated and tamoxifen-treated conditions. We have now included it in **Supplementary Fig. 1d**.

3. How many of the genes overexpressed by p42 in Fig. 1A are downregulated in Fig. 1C following p42 deletion via Cre-ERT2?

To address this question, we compared the differentially expressed genes in HPC-7 (**Fig. 1a**) and progenitor cells (**Fig. 1c**) using the same criteria used throughout Figure 1 ($\log_2FC > |1|$ and $\text{adj.P} < 0.05$). Specifically, the reviewer is asking about the genes over-expressed by p42 in **Fig. 1a**, which, according to our nomenclature, correspond to the "downregulated" genes in HPC-7 (as the comparison is p30 vs p42). Similarly, the genes downregulated in **Fig. 1c** correspond to those more highly expressed in WT than in p30-expressing cells. As shown in **Reviewer Table 1**, the vast majority of genes differentially expressed in both systems change in the same direction (indicated by grey-shaded boxes). Specifically, 202 genes are downregulated in both systems, while only 5 and 11 genes show changes in opposite directions. Please note that the total number of deregulated genes in this overlap analysis is lower than in Fig. 1 due to the difference in the total number of genes which is not the same in each comparison. Since we applied stringent criteria (considering only differentially expressed genes with \log_2FC greater than 1 or less than -1), many genes fall in the "unchanged" category. To explore how the correlation would appear without these criteria, we plotted the \log_2FC of all genes without applying any cutoff. As can be seen in **Reviewer Fig. 5**, there is a clear positive correlation between the two systems. Overall, these data points to a common transcriptional effect of p30 and p42 in HPC-7 and primary bone marrow progenitors.

		HPC-7 p30 vs p42			Total
		UP	DOWN	Unchanged	
Progenitors p30 vs WT	UP	142	5	242	389
	DOWN	11	202	137	350
	Unchanged	435	520	12459	13414

Reviewer Table 1. Overlap between HPC-7 and progenitors. UP: $\text{adj.P} < 0.05$ & $\log_2FC > 1$. DOWN: $\text{adj.P} < 0.05$ & $\log_2FC < -1$. Genes that did not fit into these criteria are shown as Unchanged. In grey: number of genes changing in the same direction in both systems.

Reviewer Fig. 5. Correlation between HPC-7 and progenitors. All genes in the analyses are shown and placed according to their log₂FC in progenitors (X axis) or HPC-7 (Y axis).

4. Given the concerns about tamoxifen, how would the inflammatory gene expression levels compare if the authors used an MX-Cre mouse model instead of Cre-ERT2 to delete p42 in macrophages and progenitors, as shown in Fig.s 1C-F?

We did not use an Mx-Cre mouse model because in this system Cre activation is dependent on the administration of polyinosinic-polycytidylic acid or poly(I:C), a synthetic analog of double-stranded RNA (dsRNA). Poly(I:C) is widely used to mimic viral infections in experimental settings because it activates the innate immune response by engaging TLR3 and other immune receptors, which activate multiple overlapping signaling pathways with TLR4. Poly(I:C) treatment directly affects HSCs (Essers et al Nature 2009; PMID: 19212321), and the functional effects are long-lasting (Bogeska et al. Cell Stem Cell 2022; PMID: 35858618). Therefore, using poly(I:C) to induce p42 deletion would have been a highly confounding effect for our analyses, where we compare baseline and LPS-treated cells.

5. Do inflammatory genes show significant differences when comparing data from previously published models, such as Cebpa^{p30/p30} HSCPs or GMPs versus wild-type HSCPs or GMPs?

To address this, we have re-analysed RNA-seq data of phenotypically defined GMPs from wild-type (WT) and leukemic Lp30 (p30/p30) mice (Jakobsen et al. Sci Adv 2019; PMID: 31309149) and conducted differential expression analysis. To ask if inflammatory genes are deregulated, we have performed GSEA, which shows that both the gene set 'Hallmark inflammatory response' and the AML inflammatory module shown in Fig. 1j are significantly

downregulated in p30/p30 GMPs (**Reviewer Fig. 6a**). A gene ontology enrichment analysis among downregulated genes in p30/p30 GMPs compared to WT GMPs shows a clear enrichment of inflammatory terms among the top ten most significant terms (**Reviewer Fig. 6b**). These data indicate that the reduced inflammatory gene expression in p30-expressing cells is not exclusive of our experimental systems and, importantly, that it is not a consequence of ERT2-fusion proteins or the use of tamoxifen.

Reviewer Fig. 6. Inflammatory response in previously published data. a, GSEA of hallmark inflammatory response genes and the AML inflammatory module in Lp30 vs WT GMPs **b**, Top 10 GO terms enriched in downregulated genes in Lp30 GMPs.

6. The gene expression findings in KO-52 cells following p42 overexpression stand in contrast with the effects observed in HPC-7 cells and the mouse model, as shown in Fig. 1A, C, and E. In the latter models, p42 overexpression resulted in a balanced number of upregulated and downregulated genes. However, in Fig. 2E, p42 overexpression predominantly upregulates genes. Could the authors explain this discrepancy and why p42 overexpression in Fig. 2E leads to mainly upregulated genes, while in Fig. 1, it shows a more balanced impact on gene expression?

As suggested by this reviewer in the following comment (number 7), we have repeated the entire experiment using external RNA spike-ins for normalization. With this approach, we identified 1,089 upregulated genes and 660 downregulated genes (**Fig. 2e**), which is a more balanced number of up- and downregulated genes compared to the experiment without spike-in normalization. Although the bias toward upregulated genes persists, we think it is expected because we are comparing cells overexpressing p42 with cells lacking any overexpression. Given the largely activatory role of CEBPA (Christou-Kent et al. Cell Rep 2023; PMID: 37516962), increased transcription of hundreds of genes upon overexpression is anticipated.

In contrast, in Fig. 1, we compared either two overexpression conditions (Fig. 1a) or primary cells without overexpression (Fig. 1c and 1e). We have updated panels e–k in **Fig. 2** and **Supplementary Fig. 3e–f** to reflect the new spike-in–normalized data.

7. Also related to the previous point, in KO-52 cells overexpressing p42, more than 2,600 genes were upregulated, as shown in Fig. 2E, while only a small fraction of genes were downregulated. This suggests a significant skew in the data, with some genes showing more than a 10-fold upregulation on a log₂ scale (which translates to over 1,000-fold changes on gene expression on a linear scale, a rather unphysiological result). To address this apparent bias in expression patterns, the authors should consider running external RNA spike-in controls if they have not already done so.

We thank the reviewer for this suggestion and, as explained in the previous point, we have now included RNA spike-in controls for normalization following the reviewer's indication. The new data not only has a more balanced number of up- and downregulated genes, but also more nuanced fold-change values, ranging from 1- to 6-fold on a log₂ scale. These values are consistent with published changes in gene expression upon CEBPA overexpression in other cellular systems (Christou-Kent et al., *Cell Reports*, 2023; PMID: 37516962).

8. Given the substantial transcriptional differences induced by p42 overexpression in KO-52 cells (as shown in Fig. 2E), the changes in inflammatory gene expression observed with p42 overexpression in Fig. 2G seem relatively minor. If normalized to the empty vector (EV) control, the fold changes caused by p42 would likely be minimal compared to the over 2600 genes that are strongly upregulated by p42, as shown in Fig. 2E. These findings suggest that inflammatory genes are only weakly affected by p42 overexpression in the absence of a tamoxifen-inducible model.

First, we would like to clarify that the normalized counts on the Y-axis of Fig. 2g are displayed on a logarithmic scale. We apologize for the oversight and have now updated the axis and figure legend in the revised manuscript. To address the reviewer's point, we have generated a volcano plot that exclusively shows inflammatory genes (the same gene set used in Fig. 2G). As shown in **Reviewer Fig. 7a**, the vast majority of deregulated inflammatory genes are upregulated ($\log_2FC > 1$ and $\text{adj.P} < 0.05$), with a broad range of fold changes, similar to the fold-change distribution observed for other upregulated genes in Fig. 2e. Next, to determine whether inflammatory genes are particularly deregulated or if any random gene set would exhibit a similar fraction of deregulated genes and fold-change distribution, we generated random gene sets of the same size as the inflammatory gene set (considering only genes

expressed in our RNA-seq data). We then estimated how many genes in the random gene sets are upregulated in our dataset. After generating 100K random gene sets, we found that it is extremely unlikely that the inflammatory gene set is upregulated by chance (p-value = $5.9e-37$, **Reviewer Fig. 7b**).

Reviewer Fig.7. Gene expression changes of inflammatory genes. a, Volcano plot of inflammatory genes comparing KO-52 transfected with p42 vs empty vector (EV). **b**, Permutation test counting the number of up-regulated genes from random gene sets with the same size as the number of expressed inflammatory genes ($n=100000$). Red dashed line indicates the number of up-regulated genes within the inflammatory gene set (46 genes).

9. While KO-52 is a CEBPA mutant case, plasmid-based overexpression of transgenes can lead to unnaturally high levels that may activate various stress pathways. Therefore, proper controls should include the overexpression of p30 in KO-52 cells. Specifically, what are the expression changes when p30 is overexpressed in KO-52 cells compared to empty vector (EV) controls, particularly regarding AP-1 family members and inflammatory genes? The authors should include analogous experiments to those shown in Fig. 2E-2I, but with p30 overexpression in KO-52 cells. Do the inflammatory genes highlighted in Fig. 2H remain low or become further downregulated when p30 is overexpressed in KO-52 cells, as assessed using GSEA approaches?

Following the reviewer's suggestion, we overexpressed p30 in KO-52 cells and performed RNA-seq, then compared these cells with p42-overexpressing KO-52 cells. We identified 582 upregulated and 528 downregulated genes ($\text{adj.P} < 0.05$; $\log_2\text{FC} > |1|$). Importantly, the top inflammatory GO terms enriched in KO-52 p42 compared to EV were also significantly enriched when comparing to p30-overexpressing KO-52 cells (**Supplementary Fig. 2d-e**). Consistently, when we repeated the GSEA analysis on the AML inflammation module of Fig. 2h, we observed a significant and marked upregulation ($\text{NES} = 2.462$; $\text{FDR} < 0.05$; **Supplementary Fig. 2f**), closely matching the results obtained using EV-transduced cells as

a control. Finally, we examined the expression of AP-1 members and found a pattern of changes similar to that seen in Fig. 2 with EV controls (**Supplementary Fig. 2g**). Overall, these findings indicate that p42 overexpression partially rescues inflammatory gene expression, and this effect is consistent whether we compare against EV-transduced controls or p30-transduced cells.

10. The inflammatory AML module, downregulated in CEBPAbi/NT-mutated AML, was upregulated following p42 transfection (Fig. 2H). How are these inflammatory genes regulated when p30 is overexpressed?

As explained in the previous point, the AML inflammatory module is upregulated in KO-52 p42 cells compared to KO-52 p30 cells (NES = 2.462). When comparing p30 cells to EV cells, we also observe upregulation of these transcripts, although it is much milder (NES = 1.921) than when p42 cells are compared to EV (NES = 2.611). These findings indicate that overexpression of p30 (in KO-52 cells, which already endogenously express p30) does enhance inflammatory gene expression, but to a lesser degree than p42. This result aligns with our general conclusion that p42 induces higher expression of inflammatory transcripts compared to p30.

11. Does p42 overexpression in KO-52 restore LPS response compared to p30 overexpressing KO-52 cells?

To address this, we stimulated with LPS for 2 hours the p30-overexpressing KO-52 cells explained above. As expected, we observe that LPS-response genes in p42-expressing cells are more expressed than in p30 and EV cells. In line with the previous comment, p30 overexpression did promote a moderate upregulation compared to EV, although not to the same extent as p42 (**Supplementary Fig. 3f**).

12. What are the gene ontology results for the 17% of overexpressed genes in p30 HPC-7 cells (Fig. 3D) and p30 macrophages (Fig. 3G) after LPS induction? Additionally, how many of these genes overlap between HPC-7 cells and macrophages?

We conducted gene ontology analysis on these two sets of genes, which are LPS-inducible genes that are more induced in p30- than in p42-expressing cells. The results revealed a variety of GO terms: for HPC-7 cells, the terms are primarily related to nuclear transport and cellular localization, while for macrophages, they are more associated with transcriptional regulation, response to GM-CSF, and DNA damage response (**Reviewer Fig. 8a**). Consistent

with these findings, there is minimal overlap between the two gene sets (**Reviewer Fig. 8b**). While these data may provide interesting insights, we have not explored them further at this stage.

Reviewer Fig. 8. Upregulated genes in HPC-7 cells and macrophages. a, Top 10 GO terms enriched in LPS-inducible genes that are more induced in p30- than in p42-expressing cells, HPC-7 (left) and macrophages (right) at 2h LPS. **b**, Overlap between upregulated LPS-activated genes in p30 HPC-7 and p30 macrophages after 2h LPS.

13. What are the GSEA scores for the Hallmark Inflammatory Response (used in Fig. 2A) and the Inflammation Module (used in Fig. 2H) when comparing p30- versus p42-expressing HPC-7 cells after LPS stimulation at 2 and 12 hours?

The GSEA scores for both gene sets are all negative at both 2 hours (-1.854 and -2.141) and 12 hours (-1.821 and -2.104) post-LPS treatment (**Reviewer Fig. 9**).

Reviewer Fig. 9. Inflammatory genes at 2h and 12h of LPS. GSEAs of Hallmark inflammatory response and the AML inflammation module at 2h and 12h of LPS treatment in p30 versus p42 HPC-7 cells.

14. The authors state that remarkably, a significant overlap in LPS-responsive genes was observed (Fig. 3H). However, only 30 overlapping genes were identified, while nearly 2,000 genes are differentially regulated between macrophages and HPCs, and about 500 more in KO-52 cells. This limited overlap does not strongly suggest a conserved inflammatory response. Additionally, it is unclear how many of these 30 genes might be detected by random chance. The authors should include a false discovery rate corrected p-value to determine if these 30 genes are statistically significant.

Since a hypergeometric test or Fisher's exact test is typically suited for calculating the significance of overlap between two groups but not more, we performed a permutation test to calculate the p-value. We randomly selected groups of expressed genes of the same size for each cellular model and calculated the number of overlaps. This process was repeated $n=100,000$ times. The p-value was derived by comparing the observed overlap with the normal distribution of overlaps from these random sets.

However, we understand the reviewer's criticism and agree that the fact that there is a 30-gene significant overlap between the three systems does not add much value to the manuscript, and we realise that this may even be misleading as we only wanted to make the point that the three cellular systems show a significant degree of consistency among them. Therefore, to avoid any confusion, we have decided to remove this Venn diagram and associated genome browser panel from the figure.

15. While CXCL10 and potentially GPR84 among the conserved 30 genes are associated with inflammation, a brief review of Sod2 and Serpine1 suggests only a weak connection to inflammation, if any. Can the authors explain how relevant these 30 genes are to the immune phenotype?

Following the reviewer's previous point, we understand the comment and as a result have removed the overlap analysis from the figure, as it did not have sufficient conceptual value to understand the main conclusions of our study.

16. It is puzzling that the authors do not present the expression levels of AP-1-related genes in the LPS-treated experiments shown in Fig. 3, given that this is central to the study. What are the expression changes of AP-1-related genes in p30 and p42 HPC-7 cells following LPS stimulation at 2 and 12 hours?

We thank the reviewer for this suggestion. We have now added a heatmap showing the fold-change in expression between p30 and p42 for the main AP-1 members at 0h, 2h, and 12h (**Supplementary Fig. 3b**). The trend at 2h and at 12h is very similar to the one at 0h, and it shows that the FOS subfamily is the most downregulated, and only Atf3 and Batf3 show upregulation.

17. The LPS-stimulated inflammatory response appears to be relatively short-lived, with no significant differences observed between p42 and p30 HPC-7 cells after 12 hours (Fig. 3A-C). How do the authors reconcile this result with the finding that thousands of genes, including inflammatory genes, are downregulated in p30-expressing HPC-7 cells even after 24 hours, as shown in Fig. 1?

The results in Fig. 1 show changes in gene expression at baseline, prior to any LPS stimulation. The 24-hour time point refers to the tamoxifen induction of nuclear translocation of the protein in the ERT2 fusion system (**Fig 1a-b**), but not in primary cells (**Fig 1 c-d, e-f**). Our interpretation of the results is that the transcriptional changes at baseline at least partially explain the deficient LPS-activation, as some of the upstream regulators of early LPS-response genes (e.g. AP-1 factors) are downregulated before LPS treatment (**Fig. 1h-i**). The late response genes, however, are mainly regulated by interferon transcription factors (IRFs, STAT1 and STAT2, **Fig. 3a**), which are not majorly affected at baseline (**Fig. 1h-i**).

18. How many genes from the p30-downregulated set shown in Fig. 1A are differentially

regulated by LPS stimulation, both during early and late activation phases? Additionally, how do the p30-upregulated genes in Fig. 1A compare in terms of differential regulation by LPS?

To visualize this, we generated volcano plots for the deregulated gene sets shown in Fig. 1A. Most p30-downregulated genes remain downregulated at 2h and 12h, with only three genes becoming upregulated (**Reviewer Fig. 10**). We performed the same analysis for the upregulated genes, reaching the same conclusion (**Reviewer Fig. 10**).

Reviewer Fig. 10. p30 deregulated genes in LPS stimulation. Volcano plots showing how deregulated genes in HPC-7 p30 vs p42 at baseline behave at 2h and 12h of LPS stimulation.

19. How do the authors reconcile the absence of differentially expressed genes (DEGs) between p30 and p42 after 12 hours (Fig. 3) with the significant changes in chemokines observed even after 16 hours (Fig. 4)?

The reviewer brings up an important point. As can be seen in Fig. 4a, most cytokine-encoding genes are markedly downregulated at 2 hours, and many still are – but less so – at 12 hours. We think that changes in expression at early time-points, such as 2 hours, may be detected at the protein level much later. First, because there is a necessary time gap between gene transcription, protein translation and protein secretion to the media. Second, because the half-lives of cytokines can be very varied, and for some cases these extend up to 15 hours

(Kuribayashi, 2018 PMID: 29937915). Indeed, gene and protein expression changes are often discordant (review: Liu et al. *Cell* 2016; PMID: 27104977). Previous work in macrophages and dendritic cells has shown that early changes in LPS inducible gene expression are often represented at later time points in the proteome (Eichelbaum 2014 PMID: 24396086 and Jovanovic 2016 PMID: 25745177). Therefore, changes in gene expression at early time points following LPS exposure in HPC-7 cells, are likely also represented in the proteome at later time points.

20. Why is CXCL10 absent in Fig. 4C and 4D after LPS stimulation, despite being identified in Fig. 3H as one of the most consistently downregulated genes by p30 following LPS activation?

We were unable to detect CXCL10 among the identified proteins using LC/MS. However, we also performed a targeted quantification of secreted proteins using an antibody-based approach, which is better suited for detecting secreted proteins that may be present at low abundance in the supernatant. Using this method, we identified CXCL10, which was markedly downregulated in the supernatant of p30-expressing cells (**Supplementary Fig. 4b**). The low abundance of CXCL10 in the secretome could be attributed to several factors, including timing, as raised by the reviewer in the previous question. It is possible that CXCL10 secretion may not have peaked at the time of our analysis. Further studies would be needed to determine the precise secretion dynamics of this protein.

21. The authors concluded that fos/AP-1 is the main regulator for the inflammatory response. How come that CXCL10 (which they identified among the 30 conserved inflammatory genes in Fig 3H) is not upregulated when FOS is transduced in KO-52 cells in Fig 6H?

We have repeated the qPCR experiments with additional replicates to determine whether CXCL10 is significantly upregulated, but the results indicate that it is not. However, it is still possible that the high Ct values (resulting from low expression levels) prevent small changes from being accurately detected. Altogether, based on the data available, we conclude that CXCL10 upregulation by p42 is not FOS-dependent. As explained in the text, our data indicate that FOS is a key mediator of the p42-dependent inflammatory response and helps to explain the differences between p30 and p42. However, we do not claim that FOS is the sole regulator of this response, which may include other AP-1 family factors, the CEBP proteins themselves, or additional indirect regulatory mechanisms. This is something we are further investigating.

22. The authors hypothesized that incomplete activation of inflammatory genes in p30-

expressing cells might mitigate the adverse effects of prolonged LPS exposure. While chronic inflammation is known to impair hematopoietic stem cell (HSC) function, this does not necessarily translate to acute myeloid leukemia (AML). Various studies suggest that inflammation in the bone marrow can promote hematological malignancies. Furthermore, the effect of LPS impairment shown in Fig. 4F is minimal and not clearly discernible due to the unfavorable y-axis break, which complicates proper interpretation. It appears that non-LPS-treated p30-overexpressing HPC-7 cells exhibit a 4-5-fold higher cell count compared to non-LPS-treated p42-transduced HPC-7 cells at day 8. Similarly, LPS treatment results in a 4-5-fold increase in cell count for p30-overexpressing cells compared to p42 cells. The impairment in p42 cells due to LPS is minimal compared to the advantage conferred by p30, regardless of LPS stimulation (Fig. 4F). Given the unfavorable y-axis break in Fig. 4F—specifically between p42 untreated and p42 LPS conditions—the authors should remove this break and present the data on a full linear scale to facilitate accurate interpretation. Overall, these results do not strongly support the notion that inflammation is the primary driver behind the competitive advantage of p30 over p42. Additional cell models, ideally primary samples, would be needed to support their hypothesis more convincingly.

We thank the reviewer for suggesting a change in the Y-axis of Fig. 4f. We have tried to remove the axis break, but the consequence is that the differences in the earlier time-point cannot be appreciated. To solve this, we have kept the broken Y axis but adding dotted lines, which indicate that the break does not separate p42 UT and p42 LPS (**Fig. 4f**). We hope that the reviewer agrees with our decision.

To further show the detrimental effects of LPS in other models, and in response to another reviewer, we have performed additional experiments in the CEBPA-mutant AML cell line KO-52. First, we assessed the colony-forming capacity of KO-52 cells overexpressing either wild-type p42 (which they do not endogenously express) or p30 (as a control) in methylcellulose. These cells were plated in methylcellulose supplemented with LPS, and colonies were counted 14 days later. While LPS only minimally reduced the number of colonies formed by p30-expressing cells, it significantly reduced the number of colonies formed by p42-expressing cells (**Supplementary Fig. 4e**). This finding is consistent with the reduced colony formation observed in mouse primary progenitors (**Fig. 4e**) and the reduced cell numbers in HPC-7 cells exposed to LPS (**Fig. 4g**), indicating that these observations are recapitulated in an AML cellular model.

Next, to assess the contribution of FOS to cellular fitness in response to inflammatory signals, we plated FOS-overexpressing KO-52 cells in LPS-supplemented methylcellulose. LPS

treatment caused a more pronounced reduction in the number of colonies formed by FOS-expressing cells compared to control cells (**Supplementary Fig. 6k-n**). Overall, these experiments indicate that the partial rescue of inflammatory gene expression by FOS leads to a greater fitness reduction in FOS-expressing cells under inflammatory conditions.

These additional experiments support our initial observations. However, we would like to clarify that we do not claim that inflammation is the main nor the only driver of competitive advantage of CEBPA-mutant cells. We believe that this advantage is likely multi-factorial, and certainly other cellular processes altered by CEBPA mutations contribute to their clonal advantage (reviewed in PMID: 31867767 and PMID: 28179278). The pleiotropic nature of cancer mutations is increasingly recognized (PMID: 32605290; PMID: 33116132, PMID: 33518400), affecting complex molecular networks that control a vast range of processes, including proliferation, differentiation, metabolic state, or the response to the extracellular environment, among others. This diversity of effects is especially true for mutations in transcriptional regulators and chromatin modifiers, which alter the expression of hundreds of genes and affect multiple cellular functions—underscoring the need to consider their broad-reaching impacts when investigating cancer biology.

In AML, inflammatory signals are particularly relevant, given the inflammatory nature of myeloid cells. Indeed, evidence is growing that these signals can affect the clonal growth of various mutations, even if they are not necessarily the main force behind clonal competition (e.g. PMID: 3766699; PMID: 29195897; PMID: 33743191). As the reviewer rightly points out, multiple examples show that inflammation can benefit certain AML mutations, underscoring the importance of investigating how each AML subtype responds to inflammatory cues. This is critical because the tumor microenvironment is an evolving niche in which the same mutant cells may respond differently as conditions change, and some mutations may be especially advantageous in adapting to these shifts. This information may be especially relevant for understanding and predicting responses to existing and emerging therapies.

Overall, we show an impaired cell growth of p42-expressing HPC-7 when exposed to LPS and we observe reduced colony-forming capacity of p42 progenitor cells and in an AML model with p42 rescue. Therefore, we are confident that LPS further diminishes the competitive advantage of p42-expressing cells relative to p30-expressing cells, adding to the other factors that drive p30's selective advantage.

Mutant CEBPA promotes tolerance to inflammatory stress through deficient AP-1 activation

REVIEWERS' COMMENTS

Reviewer #1 (Remarks to the Author):

The authors did an great job to revise the manuscript and to address the reviewers' comments. I am fully satisfied with the revision.

We thank the reviewer and appreciate the positive response and finding our revision satisfactory.

Reviewer #2 (Remarks to the Author):

I thank the Authors for the substantive amount of work performed in response to my comments, which I consider to have been sufficiently addressed.

If editorially possible, I would recommend bringing the summary diagram currently in Supplementary Fig. 8 into the main Figures. I also suggest including some of the insightful discussion by the Authors of the Minor Point 2 I raised, as it contextualises their findings in respect of time-dependent cellular evolution of CEBPA mutant leukaemia.

We thank the reviewer for the positive response and finding the revision satisfactory. Following the reviewer's suggestion, we have moved the Supplementary Figure 8 to main Figure 8. We have also expanded the discussion to include some of the ideas discussed in the previous revision regarding the evolution of CEBPA mutant leukemia (lines 582-588).

Reviewer #3 (Remarks to the Author):

The authors have conducted a series of new experiments, performed re-analyses, and made significant improvements to their manuscript. They have addressed many of the previous concerns/questions.

The remaining points are as follows: the authors decided to remove the initial Venn diagram and the associated figures related to points 14/15. However, it remains important that the authors discuss the lack of a conserved inflammatory response across the systems they used. Similarly, points 17, 19, and 20 could be briefly included in the discussion section.

The authors did not explicitly mention where each of the reviewer figures has been incorporated into the manuscript or the reasons for not including them. It would be helpful for the reader if these figures were included as supplementary material.

Aside from these points, I have no further comments or questions.

We thank the reviewer for the positive response to our revised work. Following the reviewer's suggestion, we have added an explanation in the discussion section about the lack of a large conserved inflammatory gene signature across systems as well as the how baseline changes

affect the LPS activation (lines 561-566). Also following the reviewer's suggestion, we have added Reviewer Table 1, Reviewer Fig. 7 and Reviewer Fig. 10 as Supplementary Table 1, Supplementary Fig 2d-e, and Supplementary Fig 3b, respectively. We agree with the reviewer that this additions will be helpful to follow the narrative of the manuscript.